# THE UNSEEN FRONTIER: PUSHING THE LIMITS OF LLM SPARSITY WITH SURROGATE-FREE ADMM

**Kwanhee Lee[1], Hyeondo Jang[1], Dongyeop Lee[1], Dan Alistarh[2], Namhoon Lee[1]**
[1]POSTECH    [2]ISTA
{kwanhee.lee,hyeondo.jang,dongyeop.lee2,namhoon.lee}@postech.ac.kr,
dan.alistarh@ist.ac.at

## ABSTRACT

Neural network pruning is a promising technique to mitigate the excessive computational and memory requirements of large language models (LLMs). Despite its promise, however, progress in this area has diminished, as conventional methods are seemingly unable to surpass moderate sparsity levels (50-60%) without severely degrading model accuracy. This work breaks through the current impasse, presenting a principled and effective method called ELSA, which achieves extreme sparsity levels of up to 90% while retaining high model fidelity. This is done by identifying several limitations in current practice, all of which can be traced back to their reliance on a surrogate objective formulation. ELSA tackles this issue directly and effectively via standard and well-established constrained optimization techniques based on ADMM. Our extensive experiments across a wide range of models and scales show that ELSA achieves substantial improvements over existing methods; *e.g.*, it achieves $7.8\times$ less perplexity than the best existing method on LLaMA-2-7B at 90% sparsity. Moreover, we show that ELSA remains stable even at extreme sparsity (e.g., 95%), yielding up to $\times 3.98$ inference speedup and $\times 7.80$ memory compression over its dense counterpart. We also present ELSA$_L$, a quantized variant that scales to extremely large models (27B), and establish its theoretical convergence guarantees. These results highlight meaningful progress in advancing the frontier of LLM sparsity, while promising that significant opportunities for further advancement may remain in directions that have so far attracted limited exploration.

## 1 INTRODUCTION

Large language models (LLMs) have become indispensable tools across various fields, from creative industries to scientific research, but their immense size incurs a tremendous amount of memory, computation, and energy consumption, posing a significant challenge to their widespread deployment (Kaplan et al., 2020; Bommasani, 2021; Faiz et al., 2024). Neural network pruning can offer a viable solution to this problem by removing redundant parameters without compromising performance (LeCun et al., 1989; Han et al., 2015; Hoefler et al., 2021). Indeed, the research community has responded to this challenge with a surge of innovative methodologies, demonstrating that LLMs can be made more compact and efficient through effective pruning techniques (Frantar & Alistarh, 2023; Sun et al., 2024; Boža, 2024; Meng et al., 2024; Fang et al., 2024; Liu et al., 2025; Lee et al., 2025).

However, the community is witnessing a major roadblock: current methodologies are failing to push beyond a moderate level of sparsity (roughly 50-60%) without a significant decline in model performance; for instance, prior works have highlighted this limitation with rather incremental improvements at high sparsity (Meng et al., 2024; Boža, 2024; Yin et al., 2024a; Huang et al., 2025).

*Have we truly reached a plateau, or is there a path to continued progress?*

This work provides a positive answer. We demonstrate that it is possible to prune LLMs for very high sparsity levels—up to almost 90%—without significant performance degradation (see Figure 1).

The key to our success is identifying and addressing potentially critical flaws in the current practice. Specifically, the majority of existing methods relies on the principle of sequential layerwise reconstruction error minimization, an approach proven effective in memory-constrained environments.

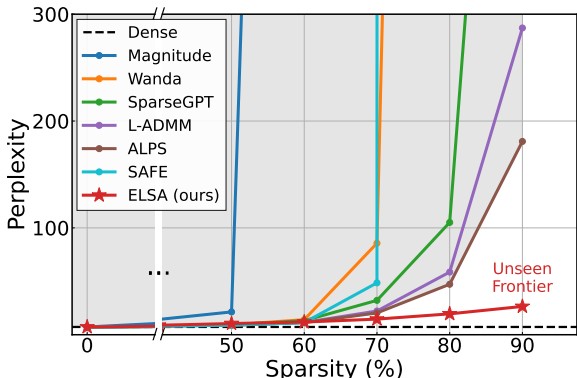

Figure 1: Perplexity (↓) vs. Sparsity (↑) curves for different pruning methods; it is measured on the C4 dataset for pruned LLaMA-2-7B models. While existing methods start to fail as sparsity increases, our approach (ELSA) stays stable without losing much performance, revealing the unseen frontier. Previously it was considered nearly impossible to achieve such high sparsity for LLMs or go beyond the "sparsity wall" formed around 50-60% sparsity levels. The same trend is observed consistently across different architectures and scales as we will show in Section 5–EXPERIMENTS.

However, this approach is inherently prone to propagating compounding errors while enforcing unnecessarily strong conditions and, in fact, seeks only local solutions by design based on a surrogate objective (Shin et al., 2024; Bai et al., 2024; Huang et al., 2025). On the other hand, we suggest finding more globally optimal solutions directly by formulating a sparsity-constrained optimization problem and developing a robust solver as a whole.

We show that our approach can be applied to a wide range of LLM models and scales from 125M to 13B number of parameters. Across this range, ELSA remains stable in the highly sparse regime, whereas competing methods frequently collapse with order-of-magnitude perplexity blow-ups. Importantly, these gains translate into practical benefits: we obtain up to $\times 7.80$ memory reduction and $\times 3.98$ inference speedup without degrading model usability. We provide a flexible implementation as well, which incorporates memory-efficient designs including quantized optimizer states and enables pruning even for 27B-parameter models with 55% lower memory footprint, demonstrating extended potential at scale. Based on classic optimization theory, we also provide a convergence guarantee for our solver to ensure theoretical soundness alongside empirical findings.

The full extent of its limits is not yet fully understood. However, our work clearly demonstrates significant potential for further advancements in LLM pruning. We believe that this finding calls for a renewed focus on alternative strategies that more faithfully preserve model fidelity, which could include better ways to exchange efficiency for performance, providing practitioners with a wider range of options.

## 2 PROBLEM STATEMENT

The long-standing research of neural network pruning, aimed at enhancing the efficiency of large models (LeCun et al., 1989; Han et al., 2015), has recently made significant progress in its application to LLMs (Frantar & Alistarh, 2023; Sun et al., 2024; Boža, 2024; Liu et al., 2025). While effective, these methods decline sharply and fail to maintain performance beyond a moderate level of sparsity around 50-60%. For example, the recent study of Zhang et al. (2024) to evaluate these methods report that their performance begins to collapse after 70% sparsity. This deterioration is also evident in other recent works that, notwithstanding the relative advantage over existing methods, the majority still suffer from severely degraded performance in high-sparsity regimes, with perplexity often increased more than an order of magnitude (Boža, 2024; Meng et al., 2024). In fact, this stands in stark contrast to historical precedents, where extreme sparsity of say 90% or higher was commonly achieved (Frankle & Carbin, 2019; Lee et al., 2019). Consequently, researchers has begun to theorize the underlying causes, attributing the failure to compounding layer-wise errors and the explosion of reconstruction error (Shin et al., 2024; Huang et al., 2025).

These findings have collectively fostered a narrative that achieving high sparsity in language models is an illusional goal. We argue, however, that this "sparsity wall" is perhaps not an inherent limitation but rather an artifact of ill-defined problem formulation.

To analyze, let us begin by showing that pruning can be formulated most generally as a constrained optimization problem as follows:

$$x^\star = \arg\min \ f(x) \quad \text{subject to} \quad \|x\|_0 \leqslant k \tag{1}$$

where $x \in \mathbb{R}^d$ refers to the optimization variable (*i.e.*, parameters of a neural network), $f$ denotes the minimization objective (*e.g.*, cross-entropy loss for next token prediction), and $k$ is the number of parameters to preserve after pruning. *I.e.*, the successful processing of (1) will yield a solution $x^\star$ that is sparse and keeps prediction performance.

However, the majority of LLM pruning methods takes an approach of the following form:

$$x^\star = \{x_i^\star \ \text{for} \ i = 1, \ldots, L\} \quad \text{where} \quad x_i^\star = \arg\min \ \tilde{f}(x_i) \quad \text{subject to} \quad \|x_i\|_0 \leqslant k_i \tag{2}$$

where $L$ refers to the number of some modularized parts of the network model–most typically layers–and $\tilde{f}$ denotes a module-wise surrogate objective that measures reconstruction error; precisely, the reconstruction error here is defined to be

$$\tilde{f} := \mathbb{E}_\mathcal{D} \|\bar{x}_i^\top g(x_{i-1}; \mathcal{D}) - x_i^\top g(x_{i-1}; \mathcal{D})\|^2 \tag{3}$$

where $g(x_{i-1}; \cdot)$ and $\bar{x}$ denote the activations of the previous layer and the $i$-th layer of the pre-trained dense model, respectively, and $\mathcal{D}$ refers to some calibration data. Thus, the model is split into submodels, and each submodel is pruned so as to match or reconstruct the predictions of the dense counterpart on some data, sequentially until the last submodel. The solution is then obtained by simply stacking these sparse submodels.

We posit that this approach (2), so-called layer-wise reconstruction error minimization, introduces non-trivial and potentially critical limitations. Specifically, we highlight three potential pitfalls: (i) errors from approximate layer-wise solutions, (ii) suboptimality in model-wide reconstruction, and (iii) the surrogacy in the objective. We elaborate these as below.

First of all, it is hard to solve (2) exactly without errors, in other words, the distance (3) cannot be zero realistically. This is due to the high cost of exactly solving sparse linear regression (Natarajan, 1995). In fact, this leads to layer-wise solvers relying on saliency-based heuristics to find approximate solutions (Frantar & Alistarh, 2023; Sun et al., 2024; Meng et al., 2024). Without zero layer-wise reconstruction errors, even small errors from each layer can compound into large overall errors, which has been observed to pose non-trivial harm to performance (Shin et al., 2024; Huang et al., 2025).

Also, its sequential, layer-wise design is naturally restrictive, potentially introducing suboptimality. By enforcing the layer-wise features to match those of a pre-trained network, it effectively restricts the search space of the potential solutions, even though no guarantee exists that the optimal sparse model would necessarily respect this requirement. Further concern stems from its independent and sequential nature; the layers are never jointly optimized, and notably, earlier layers will remain fixed even when subsequent layers change regardless of the potential suboptimality it introduces.

Lastly—and perhaps quite fundamentally—its reliance on a surrogate objective $\tilde{f}$ implies that one cannot expect to obtain a solution on (1) even after perfectly solving (2). This stands in direct opposition to the underlying goal of achieving a perfect, zero error solution on (2), whereas, in reality, it may simply lead to overfitting, failing the true objective (1) of preserving the language modeling capabilities. We expect these core issues to act as a barrier as we seek higher sparsity levels.

## 3 METHOD

We propose ELSA (Extreme LLM sparsity via Surrogate-free ADMM) to directly solve (1). We ground our approach in optimization from both first-principle and advanced techniques in order to better ensure that (1) is properly solved while enhancing effectiveness specifically for LLMs.

### 3.1 SURROGATE-FREE LLM SPARSIFICATION VIA ADMM

We solve (1) using the alternating direction method of multipliers (ADMM, Boyd et al. (2011)), a strategy involving variable splitting to decouple the intractable sparsity constraint $\mathcal{S} = \{v \in \mathbb{R}^d \ |$

$\|v\|_0 \leqslant k\}$ from the training objective. This is done by introducing an auxiliary variable $z$ in the following manner:

$$\min_{x,z} f(x) + I_{\mathcal{S}}(z) \quad \text{s.t.} \quad x = z, \tag{4}$$

where $I_{\mathcal{S}}(z)$ is the indicator function for the set $\mathcal{S}$:

$$I_{\mathcal{S}}(z) := \begin{cases} 0 & \text{if } z \in \mathcal{S} \\ \infty & \text{otherwise.} \end{cases} \tag{5}$$

In turn, we keep $x$ constrained to be equal to $z$. This allows us to handle the model training and the sparsity satisfaction somewhat separated, making both much easier to handle.

To solve for this new formulation, the augmented Lagrangian can be used:

$$\mathcal{L}_{\lambda}(x, z, u) = f(x) + I_{\mathcal{S}}(z) + \frac{\lambda}{2}\|x - z + u\|_2^2 - \frac{\lambda}{2}\|u\|_2^2 \,, \tag{6}$$

where $\lambda$ is the hyperparameter for adjusting the strength of the proximal penalty, and $u$ is a scaled dual variable. ADMM solves this by alternating between minimizing the augmented Lagrangian over the primal variables $(x, z)$ and performing a dual ascent step on $u$. This decomposes the problem into three manageable subproblems that are iterated until convergence:

$$x^{t+1} = \arg\min_x \left( f(x) + \frac{\lambda}{2}\|x - z^t + u^t\|_2^2 \right) \,, \tag{7}$$

$$z^{t+1} = \arg\min_{z \in \mathcal{S}} \frac{\lambda}{2}\|x - z^t + u^t\|_2^2 = \Pi_{\mathcal{S}}(x^{t+1} + u^t) \,, \tag{8}$$

$$u^{t+1} = u^t + x^{t+1} - z^{t+1} \,. \tag{9}$$

The $x$-update (7) accounts for minimizing the training objective, and is iteratively minimized while $x$ is pushed closer to the sparse $z$. The $z$-update (8) can be expressed as the projection $\Pi_{\mathcal{S}}(x^{t+1} + u^t)$. Here, the objective associated with its $\mathcal{S}$ is simplified to minimizing the Euclidean distance from $x^{t+1} + u^t$, effectively replacing the complex, non-convex $f$ with a tractable, convex quadratic function. As a result, this has an exact closed-form solution computable by zeroing out the $(d - k)$-entries with the smallest magnitude (Lee et al., 2025). Finally, the scaled dual variable $u$ is updated in (9) to maximize the augmented Lagrangian via a single step of gradient ascent.

### 3.2 OBJECTIVE-AWARE PROJECTION

Closely inspecting the projection step in the $z$-update (8), one can see that the Euclidean distance is far too removed from $f$. Thus, it is reasonable to expect that the sparse parameters obtained in $z$ may differ considerably from the actual sparse optima of $f$.

This motivates us to align the projection step with $f$ by modifying its objective into the following quadratic:

$$z^{t+1} = \arg\min_{z \in \mathcal{S}} \frac{1}{2}(z - (x^{t+1} + u^t))^{\top} \mathbf{H} (z - (x^{t+1} + u^t)), \tag{10}$$

where $\mathbf{H}$ is the Hessian of $f$. Equivalently, we project in the $\mathbf{H}$ induced norm, aligning the step with the second-order geometry of $f$. Placed once again in the context of pruning research, its advantages would be akin to those of the family of approaches based on the Optimal Brain Surgeon algorithm (LeCun et al., 1989).

In practice, two approximations are introduced. We notice that the procedural simplicity in the Euclidean case stems from the objective being separable across entries. We found that using $\text{Diag}(\mathbf{H})$ allows us to retain this simplicity while still keeping the benefits by zeroing the entries with the smallest contribution to the objective rather than by their magnitudes. Also, we employ the Gauss-Newton approximation of the Hessian or the empirical Fisher information matrix $\hat{\mathbf{F}}$, which allows us to obtain a good approximation of the Hessian only by the outer products of the gradients (Martens, 2020). The results of these can be summarized into the following formula:

$$z^{t+1} = \arg\min_{z \in \mathcal{S}} \sum_{i \leqslant d} \hat{\mathbf{F}}_{ii} (z_i - (x_i^{t+1} + u_i^t))^2, \tag{11}$$

where each coordinate $i$ contributes independently to this new loss function. Luckily, the standard Adam optimizer has already made $\hat{\mathbf{F}}$ available for free via its second-moment estimates, requiring no additional cost in implementing this enhancement (Li et al., 2025). Overall, this tailors our algorithm ELSA to better adapt to the complex objective of LLMs, and in a way that incurs negligible additional cost.

### 3.3 SCALABLE ADMM VIA LOW-PRECISION STATES

We further enhance scalability by proposing ELSA-L. Here, we rely on two core operations: a quantization operation, $\mathcal{Q}$, that maps high-precision tensors to a compact low-precision representation, and a dequantization operation, $\mathcal{R}$, that rematerializes them.

Formally, for a high-precision tensor $z \in \mathbb{R}^d$, the $\mathcal{Q}$ operation produces a storable pair $(z_q, s)$ consisting of a quantized tensor and a scale:

$$\mathcal{Q}(z) \triangleq (z_q, s), \quad \text{where} \quad s = \max(|z|)/v_{\max} \text{ and } z_q = \text{round}(z/s). \tag{12}$$

Here, $v_{\max}$ is the maximum representable absolute value of the target data type (*e.g.*, 127 for signed INT8). Conversely, the $\mathcal{R}$ operation rematerializes the high-precision tensor from the stored pair:

$$\mathcal{R}(z_q, s) \triangleq s \cdot z_q. \tag{13}$$

These operations are applied in a cycle to manage the auxiliary variables. After a high-precision update yields an intermediate state, for instance $z^{t+1} = \Pi_{\mathcal{S}}(x^{t+1} + u^t)$, it is quantized for efficient storage: $(z_q^{t+1}, s^{t+1}) = \mathcal{Q}(z^{t+1})$. This transition yields substantial memory savings; for instance, storing a state in FP8 (8 bits) reduces the memory footprint by $4\times$ compared to the standard FP32 representation (32 bits). The overhead from the scale factor is negligible, as typically only a single 32-bit scale value is stored for the entire tensor. For the subsequent computation, the state is first rematerialized to high precision: $\hat{z}^{t+1} = \mathcal{R}(z_q^{t+1}, s^{t+1})$.

This quant-dequant cycle, which bridges low-precision storage with high-precision updates via a dynamic, data-aware scale, is a general and established principle in low-precision deep learning (Gholami et al., 2022). The specific definitions in (12) can be adapted for various formats, including both 8-bit integers (INT8) (Jacob et al., 2018) and modern floating-point types like FP8, representing a cornerstone of efficient numerical methods (Micikevicius et al., 2022).

However, this introduces nontrivial changes into the algorithm, and thus, the guarantees of ADMM do not automatically extend. We therefore establish a proof to demonstrate that ELSA-L, alongside with ELSA, will converge to the solution of (1) in the following section.

## 4 CONVERGENCE ANALYSIS

We establish theoretical convergence for both ELSA and ELSA-L to support their reliability in directly solving (1). Formally, we assume the following:

**Assumption 4.1.** *(Lower bounded on constraint) The function $f$ is lower bounded on $\mathcal{S}$. That is, there exists a constant $f_{\min} := \min_{a \in \mathcal{S}} f(a)$ and $f_{\min} > -\infty$.*

**Assumption 4.2.** *($\beta$-smoothness) The function $f$ is differentiable, and its gradient is $\beta$-smooth. That is, $\|\nabla f(x) - \nabla f(y)\| \leqslant \beta \|x - y\|$*

**Assumption 4.3.** *($\mu$-weak convexity) There exists a constant $\mu \geqslant 0$ such that $f$ is $\mu$-weakly convex. i.e., $f(x) + \frac{\mu}{2}\|x\|^2$ is convex.*

Also, we rely on the notion of $\lambda$-stationarity (Huang et al., 2021):

**Definition 4.4.** *($\lambda$-stationary point) We say a point $\bar{x}$ is a $\lambda$-stationary point of the optimization problem (1) if $\bar{x} \in \arg\min_{x \in \mathcal{S}} \left\| x - \left( \bar{x} - \lambda^{-1}\nabla f(\bar{x}) \right) \right\|$,*

*i.e.*, the point $\bar{x}$ cannot be locally improved using projected gradient descent with step-size $\lambda^{-1}$.

Given these, we present the convergence of ELSA and ELSA-L as follows:

**Corollary 4.5.** *(Convergence of ELSA) Suppose that Assumptions 4.1-4.3 hold. Assume further that $\lambda$ is chosen large enough so that $\lambda^{-1}\beta^2 - (\lambda - \mu)/2 < 0$. Let $(\bar{x}, \bar{z}, \bar{u})$ be a limit point of ELSA algorithm. Then $\bar{x}$ is a $\lambda$-stationary point of (1).*

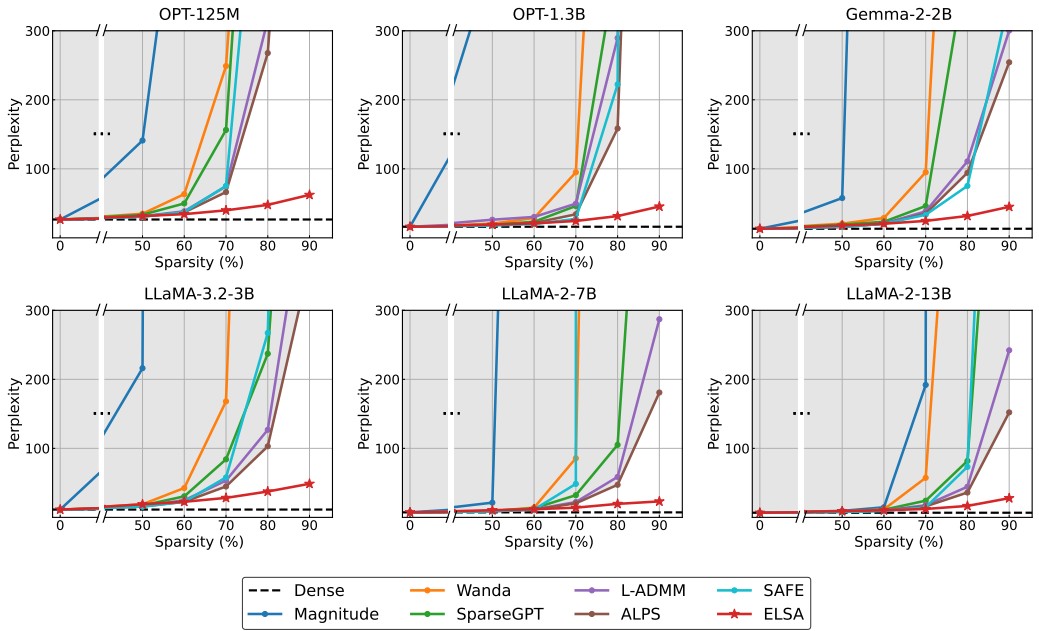

Figure 2: Perplexity vs. Sparsity plots for different models and scales. ELSA preserves much lower perplexity at high sparsity compared to other methods, consistently across a wide range of settings, showing its advantage and robustness. All numerical results are provided in Appendix C.

**Theorem 4.6.** *(Convergence of* ELSA$_{-L}$*) Suppose that Assumptions 4.1-4.3 hold. Also assume that the iterates of* ELSA$_{-L}$ *are bounded, and the constant $\lambda$ and $\gamma$ are chosen such that*

$$\frac{\beta^2}{\lambda} + \frac{\beta(\lambda + \beta)\gamma}{\lambda} + \frac{\gamma^2(\lambda + \beta)}{2} - \frac{(1-\gamma)^2(\lambda - \mu)}{2} < 0.$$

*Then, for any limit point $(\bar{x}, \bar{z}, \bar{u})$ of the iterates, $\bar{x}$ is a $\lambda$–stationary point of (1).*

This demonstrates that ELSA and ELSA$_{-L}$ converge to the stationary point of the sparsity-constrained optimization problem (1). The detailed proof for ELSA$_{-L}$ is provided in Appendix A.

## 5 EXPERIMENTS

We present a series of concrete experiments to validate ELSA in this section. Specifically, we show that ELSA (i) effectively prunes models to extreme high sparsity levels across a wide range of models and scales (Section 5.1), (ii) can further push the limits to extreme sparsity (up to 99%) (Section 5.2), (iii) can accelerate inference while reducing memory requirements (Section 5.3), and (iv) scales efficiently to large models up to 27B (Section 5.4). Lastly, we analyze the cost of ELSA compared to existing methods at (Section 5.5). We also provide extension to other sparsity patterns such as N:M semi-structured sparsity or non-uniform sparsity, and an ablation study on the choice of objective functions and generalized projection on Appendix C.

We compare ELSA to the following methods: Magnitude (Han et al., 2015), SparseGPT (Frantar & Alistarh, 2023), Wanda (Sun et al., 2024), ALPS (Meng et al., 2024), L-ADMM (Layer-wise ADMM) (Boža, 2024), SAFE (Lee et al., 2025), and SparseLLM (Bai et al., 2024). These methods are applied to four different architectures including OPT (Zhang et al., 2022), Gemma-2 (Team et al., 2024), and LLaMA-2/3 (Touvron et al., 2023; Grattafiori et al., 2024) across a wide range of scales from 125M to 27B. We report perplexity and zero-shot prediction accuracy of pruned models. All experiment settings can be found in Appendix B. The source code to reproduce results will be made available at `https://github.com/log-postech/elsa`.

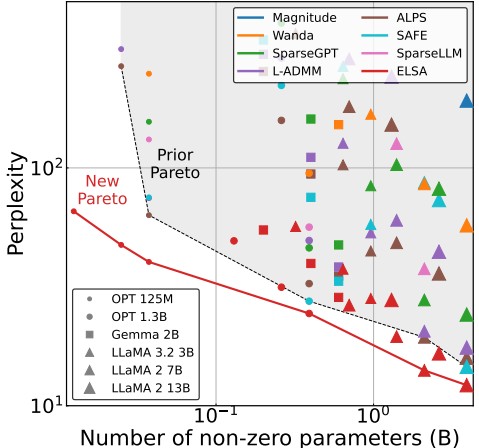

Figure 3: Pareto optimality of ELSA compared to prior works in terms of perplexity vs. number of non-zero parameters. ELSA displays its greater optimality across a broad spectrum of effective scales.

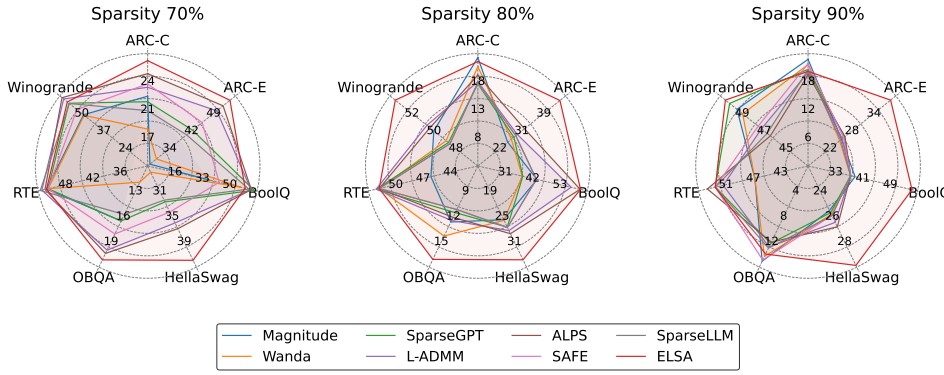

Figure 4: Zero-shot accuracy of pruned LLaMA-2-7B models. ELSA outperforms other methods for most tasks, with the performance gap widening as sparsity increases, highlighting its strong generalization capability. Full numerical results are provided in Table 11 of Appendix C.

## 5.1 MAIN RESULTS

Figure 2 reports C4 perplexity for various models across different sparsity levels from 50% to 90%. Existing methods deteriorate rapidly beyond 70%; for instance, SparseGPT on OPT-125M rises from 49.83 at 60% sparsity to over 1,000 at 80%. In contrast, ELSA remains stable, increasing only from 42.99 to 47.45 over the same range, and at 80% sparsity matches the perplexity of SparseGPT at 60%. This robustness holds across scales: on LLaMA-2-13B at 90% sparsity, ELSA achieves 27.84 perplexity, while most existing methods exceed the hundreds. Figure 3 further highlights this trend by plotting perplexity against the effective number of non-zero parameters. ELSA consistently sets the new Pareto frontier across scales, underscoring its robustness in extreme sparsity regimes.

This extends to downstream task performance, as shown in Figure 4. Each radar plot reports per-task accuracy at high sparsity (70–90%), with the enclosed area reflecting the average accuracy across tasks. At 70% sparsity, ELSA is competitive with leading methods, but a clear gap emerges as sparsity increases. From 70% to 80% sparsity, other methods lose 10–20%p accuracy on tasks such as Winogrande and ARC-E, while ELSA degrades by less than half as much. At 90%, most methods collapse, whereas ELSA retains the highest accuracy on 6 out of 7 tasks, with an average margin of 6.06%p. This demonstrates that ELSA maintains generalization far better than existing methods at high sparsity. We believe that these results collectively establish the effectiveness of ELSA for high sparsity.

Table 1: Memory savings and inference accelerations of ELSA on LLaMA-2-7B.

|  | Dense | 50% | 70% | 90% | 95% |
|---|---|---|---|---|---|
| **Latency (s)** | 1.84 | 1.37 (×1.34) | 0.95 (×1.94) | **0.72** (×**2.50**) | **0.46** (×**4.00**) |
| **Tokens/s** | 54.47 | 72.81 (×1.33) | 104.89 (×1.93) | **139.47** (×**2.56**) | **216.98** (×**3.98**) |
| **Memory (MB)** | 13596 | 8870 (×1.53) | 5603 (×2.42) | **2918** (×**4.60**) | **1743** (×**7.80**) |

## 5.2 TOWARDS EXTREME SPARSITY

To assess whether ELSA remains effective under extreme sparsity, we further evaluate LLaMA 2-7B at 95% and 99% sparsity. We additionally consider retraining after pruning with Wanda (Sun et al., 2024) for the baseline, and the experimental details can be found at Appendix B.3.

As shown in Table 2, ELSA consistently achieves substantially lower perplexity on both WikiText and C4 dataset, and the margin increases as sparsity becomes more extreme. Notably, at 99% sparsity, ELSA remains stable (55.94/40.10), whereas Wanda+full fine-tuning degrades (146.37/71.64) and Wanda+LoRA collapses (588.3/247.5). These results show that ELSA remains stable even in this extreme regime, while simple heuristics fail to preserve performance, highlighting the advantage of ELSA 's more principled approach.

Table 2: Perplexity on Wiki/C4 at extreme sparsity.

| Sparsity | Method | Wiki | C4 |
|---|---|---|---|
| 0.90 | Wanda + LoRA | 92.66 | 65.56 |
|  | Wanda + Full | 42.40 | 34.87 |
|  | ELSA | **26.97** | **23.14** |
| 0.95 | Wanda + LoRA | 371.0 | 143.0 |
|  | Wanda + Full | 84.30 | 53.62 |
|  | ELSA | **38.91** | **28.39** |
| 0.99 | Wanda + LoRA | 588.3 | 247.5 |
|  | Wanda + Full | 146.37 | 71.64 |
|  | ELSA | **55.94** | **40.10** |

## 5.3 REALIZING BENEFITS WITH EXTREME SPARSITY

Sparsity is practically meaningful only if it yields real deployment gains beyond reducing parameter count. To quantify this, we benchmark end-to-end text generation on LLaMA 2-7B using MACKO (Macko & Boža, 2025), a recent Sparse-Matrix Vector multiplication (SpMV) kernel combined with memory-efficient MACKO format, supporting an acceleration and memory savings for sparse models. Following MACKO's standard end-to-end protocol, we convert the sparse models produced by ELSA and evaluate on a single NVIDIA RTX 3090, reporting mean latency of token generation (s), throughput (tokens/s), and memory footprint (MB).

Table 1 shows that sparsity yields clear gains in both speed and memory, with improvements becoming pronounced from 70% sparsity onward. At 90% sparsity, ELSA achieves a $2.50\times$ reduction in end-to-end latency and a $2.56\times$ increase in throughput, together with a $4.60\times$ memory reduction compared to the dense baseline. Pushing further to 95% sparsity provides even stronger efficiency gains (up to $4.00\times$ latency reduction, $3.98\times$ throughput increase, and $7.80\times$ memory reduction). These results validate that the high/extreme-sparsity regime targeted by ELSA is not merely an academic frontier: it enables tangible acceleration and substantial memory savings in memory-constrained settings, particularly benefiting the decoding phase where sparse matrix–vector computation dominates.

## 5.4 SCALING TO LARGE-R MODELS

We further validate the scalability of our approach by applying $\text{ELSA}_L$ to 27B-scale (Gemma-2-27B). Specifically, we additionally employ the low-precision optimizer `adam8bit` (Dettmers et al., 2022) for $x$-update Equation (7), with $\text{ELSA}_L$, where we use (`bf16`, `fp8`) for auxiliary state $(u, z)$, respectively. This design reduces the memory footprint of required states (optimizer, auxiliary variables) by 55% compared to ELSA, enabling pruning at 27B scale under limited resources. Additional implementation details can be found in Appendix B.4.

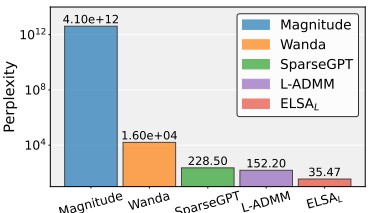

Figure 5: Perplexity of Gemma-2-27B at 90% sparsity.

Table 3: Compute cost vs. perplexity of different pruners on LLaMA-2-7B at 90% sparsity.

| Method | Wall-clock (h) | #GPUs | Wiki | C4 |
|---|---|---|---|---|
| Wanda | 0.159 | 1 | $2.0 \times 10^4$ | $1.0 \times 10^4$ |
| SparseGPT | 0.251 | 1 | 1430 | 864.5 |
| ALPS | 12.57 | 1 | 248 | 180.9 |
| Wanda+LoRA | 4.03 | 1 | 92.66 | 65.56 |
| Wanda+Full | 1.64 | 4 | 42.40 | 34.87 |
| ELSA | 1.78 | 4 | **26.97** | **23.14** |

Figure 5 demonstrates that $\text{ELSA}_{\text{L}}$ achieves the lowest perplexity among all compared methods, outperforming the strongest competing method by a factor of $4\times$, supporting our main results at scale.

## 5.5 COST ANALYSIS

To assess *effective efficiency* (i.e., quality achieved per pruning compute at fixed sparsity), we measure end-to-end pruning cost on identical NVIDIA A100-80GB GPUs and report GPU-hours together with perplexity for LLaMA-2-7B at 90% sparsity (Table 3). One-shot methods (Wanda, SparseGPT) are the cheapest (0.16/0.25 hours with single GPU) but collapse at this sparsity level, yielding unusable perplexities ($\geqslant 10^4$). The best-performing baseline ALPS improves perplexity but still remains far from usable, while requiring substantial wall-clock time (12.57 hours). Overall, these layer-wise pruners are memory-light by design, yet additional pruning compute does not reliably translate into commensurate quality gains under high sparsity.

In contrast, ELSA achieves substantially lower perplexity (26.97/23.14) with a moderate compute budget (7.12 GPU-hours), yielding a markedly better cost–quality point in this regime. Moreover, even when augmented with matched-budget retraining (Wanda+LoRA/Full), one-shot+retrain baselines do not reach comparable quality, indicating that additional compute is more effectively converted into model quality by ELSA at high sparsity.

## 6 DISCUSSION

In this work, we confront the problem of moderate sparsity in LLMs through a critical inspection into the current practice, revealing that the prevailing reliance on the sequential layer-wise reconstruction surrogate may have been constraining the path toward more extreme sparsities. This led us to develop ELSA and $\text{ELSA}_{\text{L}}$, enabling us to push the sparsity from 50–70% up to 80–90% while maintaining strong language modeling performance, where we also observe tangible deployment benefits such as inference acceleration and memory savings, and further showing that our approach remains effective even in more extreme regimes (e.g., 95–99%). Grounding on optimization principles ensures that our principle effectively solves the true LLM objective as is, while also facilitating the development of advanced techniques that are both theoretically sound and effective for sparsifying LLMs, which we believe were instrumental in attaining strong practical results.

Meanwhile, we remark on the memory demands associated with pruning LLMs. In particular, we propose to reassess the widespread assumption that, given the limitations of commodity memory, the adoption of a layer-wise surrogate strategy is difficult to circumvent. First of all, it is worth questioning whether the underlying assumption itself is too restrictive—after all, one would not typically attempt to prune an LLM without at least the resources required to run one. Also, we raise doubts about whether the layer-wise strategy provides clear memory advantages. Precisely, using the offloading technique allows one to optimize over the entire model with similar memory efficiency. In fact, quite the opposite may be the case—they do not scale well with the size of calibration data, requiring the layer activations of the entire calibration data to be stored, while a single mini-batch usually suffices the surrogate-free principle. This calls into question whether our perception of its efficiency could be somewhat inflated, requiring the need for a careful assessment of current practice and exploration of alternative strategies through a more balanced lens.

There are many promising directions to pursue for future work: (i) alternative efficiency strategies through advanced memory-efficient and derivative-free optimizers, (ii) system-level advancements in memory offloading, and (iii) extensions to advanced architecture such as Mixture-of-Experts (Mu & Lin, 2025) and multi-modal large language models (Yin et al., 2024b). To conclude, our work validates that the frontier of LLM sparsity can still be expanded by offering a concrete strategy supported by strong empirical evidence. We hope it sets the stage for future breakthroughs and innovations in new directions that have thus far received relatively limited attention.

## ACKNOWLEDGEMENTS

This work was partly supported by the Institute of Information & communications Technology Planning & Evaluation (IITP) grant funded by the Korean government (MSIT) (RS-2019-II191906, Artificial Intelligence Graduate School Program (POSTECH),RS-2022-II220959, (part2) Few-Shot learning of Causal Inference in Vision and Language for Decision Making), the National Research Foundation of Korea (NRF) grant funded by the Korean government (MSIT) ( RS-2023-00210466, RS-2025-02264052).

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

## A   PROOF OF THEOREM 4.6

Here we present the convergence proof of $\text{ELSA}_L$. Formally, we prove the convergence of the following algorithm:

---
**Algorithm 1** $\text{ELSA}_L$

---
1: **Input**: Constant $\lambda > 0$; initial points $x_0$, $u_0 \in \mathbb{R}^d$
2: **for** $r = 0, 1, 2, \ldots$ **do**
3:     **Update** $y$:    $\Pi_{\mathcal{S}}(x^t + \lambda^{-1}u^t)$
4:     **Update** $x$ by finding a point $x^{t+1}$ satisfying $\nabla f(x^{t+1}) + \mathcal{Q}[u^t + \lambda(x^{t+1} - z^{t+1})]=0$ and $\|x^{t+1} - x_\star^{t+1}\| \leqslant \gamma \min\left\{\|x^{t+1} - z^{t+1}\|, \|x^{t+1} - x^t\|\right\}$
5:     **Update** $u$:   $u^{t+1} = \mathcal{Q}[u^t + \lambda(x^{t+1} - z^{t+1})]$
6: **end for**

---

First, let us define:

$$e^t = \nabla_x \mathcal{L}(x^t, z^t, u^{t-1}) \tag{14}$$

$$= \nabla f(x^t) + u^{t-1} + \lambda(x^t - z^t) \tag{15}$$

$$= u^{t-1} + \lambda(x^t - z^t) - \mathcal{Q}[u^{t-1} + \lambda(x^t - z^t)]. \tag{16}$$

Thus, we can express the $u$ step in terms of $e^t$ as follows

$$u^{t+1} = \mathcal{Q}[u^t + \lambda(x^{t+1} - z^{t+1})] \tag{17}$$

$$= u^t + \lambda(x^{t+1} - z^{t+1}) - e^{t+1} \tag{18}$$

**Lemma A.1.** *Due to $(\lambda - \mu)$-strong convexity and $(\beta + \lambda)$-smoothness of $\mathcal{L}(\cdot, z^t, u^{t-1})$, we know that*

$$(\lambda - \mu)\|x^t - x_\star^t\| \leqslant \|e^t\| \leqslant (\lambda + \beta)\|x^t - x_\star^t\| \tag{19}$$

*Moreover, due to strong convexity we also know that:*

$$\langle e^t, x^t - x_\star^t \rangle \geqslant (\lambda - \mu)\|x^t - x_\star^t\|^2 \tag{20}$$

**Lemma A.2.** *If $\lambda \geqslant \beta$ and we also assume that the iterates $x^t$ stay bounded. Then there exists a non-negative number $\bar{D}$ s.t. $\|x^t - z^t\| \leqslant \bar{D}$. With this definition,*

$$\mathcal{L}(x^t, z^t, u^t) \geqslant f_{\min} - \gamma(\lambda + \beta)\bar{D}^2 \tag{21}$$

*Proof.* Note that

$$\mathcal{L}(x^t, z^t, u^t) = f(x^t) + \langle u^t, x^t - z^t \rangle + \frac{\lambda}{2}\|x^t - z^t\|^2 \tag{22}$$

$$= \underbrace{f(x^t) + \langle \nabla f(x^t), z^t - x^t \rangle + \frac{\lambda}{2}\|x^t - z^t\|^2}_{\geqslant f(z^t)} + \langle e^t, x^t - z^t \rangle \tag{23}$$

$$\geqslant f(z^t) - \|e^t\|\|x^t - z^t\| \tag{24}$$

$$\geqslant f_{\min} - \gamma(\lambda + \beta)\bar{D}^2 \tag{25}$$

where the last inequality is due to the assumptions and Lemma A.1. $\qquad\square$

Now let us prove sufficient decrease on $\mathcal{L}$ in each iteration.

**Lemma A.3.** *Let the assumptions of Lemma A.2 be true. Also, assume that the parameters $\lambda$ and $\gamma$ are chosen such that*

$$\frac{\beta^2}{\lambda} + \frac{\beta(\lambda + \beta)\gamma}{\lambda} + \frac{\gamma^2(\lambda + \beta)}{2} - \frac{(1 - \gamma)^2(\lambda - \mu)}{2} < 0. \tag{26}$$

*Note that $\lambda - \mu \geqslant 0$. Then, we have*

$$\lim_{r \to \infty} \|x^{t+1} - x^t\| = 0. \tag{27}$$

*Proof.* Let

$$\mathcal{L}(x^{t+1}, z^{t+1}, u^{t+1}) - \mathcal{L}(x^t, z^t, u^t) = \underbrace{\mathcal{L}(x^{t+1}, z^{t+1}, u^{t+1}) - \mathcal{L}(x^{t+1}, z^{t+1}, u^t)}_{(A)}$$
$$+ \underbrace{\mathcal{L}(x^{t+1}, z^{t+1}, u^t) - \mathcal{L}(x^t, z^t, u^t)}_{(B)}.$$

We want to show that $(A) + (B) \leqslant 0$.

$$(A) = \langle u^{t+1}, x^{t+1} - z^{t+1} \rangle - \langle u^t, x^{t+1} - z^{t+1} \rangle = \lambda^{-1}\left( \left\| u^{t+1} - u^t \right\|^2 + \langle e^{t+1}, u^{t+1} - u^t \rangle \right).$$

Using our definitions, we have

$$(A) = \lambda^{-1}\left( \|u^{t+1} - u^t\|^2 + \langle e^{t+1}, u^{t+1} - u^t \rangle \right) \tag{28}$$

$$= \lambda^{-1}\left( \|\nabla f(x^{t+1}) - \nabla f(x^t)\|^2 + \langle e^{t+1}, \nabla f(x^{t+1}) - \nabla f(x^t) \rangle \right) \tag{29}$$

$$\leqslant \lambda^{-1}\left( \|\nabla f(x^{t+1}) - \nabla f(x^t)\|^2 + \|e^{t+1}\|\|\nabla f(x^{t+1}) - \nabla f(x^t)\| \right) \tag{30}$$

$$\leqslant \lambda^{-1}\left( \beta^2 \|x^{t+1} - x^t\|^2 + \beta \|e^{t+1}\|\|x^{t+1} - x^t\| \right) \tag{31}$$

$$\leqslant \lambda^{-1}\left( \beta^2 \|x^{t+1} - x^t\|^2 + \beta(\lambda + \beta)\|x^{t+1} - x_\star^{t+1}\|\|x^{t+1} - x^t\| \right) \tag{32}$$

$$\leqslant \lambda^{-1}\left( \beta^2 \|x^{t+1} - x^t\|^2 + \beta(\lambda + \beta)\gamma\|x^{t+1} - x^t\|^2 \right) \tag{33}$$

$$= \lambda^{-1}\beta\left( \beta + (\lambda + \beta)\gamma \right)\|x^{t+1} - x^t\|^2, \tag{34}$$

where the last inequality is due to Lemma A.1 and the way $x^t$ is chosen in Algorithm 1.

On the other hand:

$$(B) = \mathcal{L}(x^{t+1}, z^{t+1}, u^t) - \mathcal{L}(x^t, z^t, u^t)$$
$$= \mathcal{L}(x^{t+1}, z^{t+1}, u^t) - \mathcal{L}(x^t, z^{t+1}, u^t) + \underbrace{\mathcal{L}(x^t, z^{t+1}, u^t) - \mathcal{L}(x^t, z^t, u^t)}_{\leqslant 0 \text{ (due to update of } y)}$$
$$\leqslant \mathcal{L}(x^{t+1}, z^{t+1}, u^t) - \mathcal{L}(x^t, z^{t+1}, u^t)$$
$$= \underbrace{\mathcal{L}(x^{t+1}, z^{t+1}, u^t) - \mathcal{L}(x_\star^{t+1}, z^{t+1}, u^t)}_{\leqslant \frac{\beta+\lambda}{2}\|x^{t+1} - x_\star^{t+1}\|^2} + \underbrace{\mathcal{L}(x_\star^{t+1}, z^{t+1}, u^t) - \mathcal{L}(x^t, z^{t+1}, u^t)}_{\leqslant -\frac{(\lambda-\mu)}{2}\|x_\star^{t+1} - x^t\|^2}$$
$$\leqslant \frac{\beta + \lambda}{2}\|x^{t+1} - x_\star^{t+1}\|^2 - \frac{(\lambda - \mu)}{2}\|x_\star^{t+1} - x^t\|^2,$$

Now note that $\|x^t - x_\star^{t+1}\| \geqslant (1 - \gamma)\|x^{t+1} - x^t\|$ and $\|x^{t+1} - x_\star^{t+1}\| \leqslant \gamma\|x^{t+1} - x^t\|$ because of the update rules of Algorithm 1. Plugging in these, we get

$$(B) \leqslant \left( \frac{\gamma^2(\lambda + \beta)}{2} - \frac{(1 - \gamma)^2(\lambda - \mu)}{2} \right)\|x^{t+1} - x^t\|^2 \tag{35}$$

Now combining the inequalities for $(A)$ and $(B)$, we have

$$\mathcal{L}(x^{t+1}, z^{t+1}, u^{t+1}) - \mathcal{L}(x^t, z^t, u^t) \tag{36}$$

$$\leqslant \underbrace{\left( \frac{\beta^2}{\lambda} + \frac{\beta(\lambda + \beta)\gamma}{\lambda} + \frac{\gamma^2(\lambda + \beta)}{2} - \frac{(1 - \gamma)^2(\lambda - \mu)}{2} \right)}_{\alpha}\|x^{t+1} - x^t\|^2 \tag{37}$$

Now for any $T$:

$$f_{\min} - \gamma(\lambda + \beta)\bar{D}^2 \leqslant \mathcal{L}(x^{T+1}, z^{T+1}, u^{T+1}) \tag{38}$$

$$= \mathcal{L}(x^0, z^0, u^0) + \sum_{t=0}^{T} \mathcal{L}(x^{t+1}, z^{t+1}, u^{t+1}) - \mathcal{L}(x^t, z^t, u^t) \tag{39}$$

$$\leqslant \alpha \sum_{t=0}^{T} \|x^{t+1} - x^t\|^2 + \mathcal{L}(x^0, z^0, u^0). \tag{40}$$

Now if the parameters are chosen appropriately such that $\alpha < 0$, then the right hand side of the above inequality is decreasing as $T$ increases, while the left hand side is constant. Therefore, we have $\lim_{T \to \infty} \sum_{t=0}^{T} \|x^{t+1} - x^t\|^2 < \infty$. Thus, $\lim_{r \to \infty} \|x^{t+1} - x^t\| = 0$. $\qquad \square$

**Theorem A.4.** *Assume that all the assumptions of Lemma A.3 is satisfied. Then, For any limit point $(\bar{x}, \bar{z}, \bar{\lambda})$ of the Algorithm 1, $\bar{x}$ is a $\lambda$-stationary solution of the problem.*

*Proof.* Consider a sub-sequence $(x^{r_t}, z^{r_t}, u^{r_t})$, for $t = 0, \cdots$ which converges to $(\bar{x}, \bar{z}, \bar{u})$. First of all due to Lemma A.3, we know that $\lim_{t \to \infty} \|x^{r_t+1} - x^{r_t}\| = 0$ and $\lim_{t \to \infty} \|x^{r_t-1} - x^{r_t}\| = 0$. Thus,

$$\lim_{t \to \infty} x^{r_t+1} = \bar{x} \ \ \& \ \ \lim_{t \to \infty} x^{r_t-1} = \bar{x} \tag{41}$$

Moreover, due to the updates of the algorithm

$$\lim_{t \to \infty} \|x^{r_t+1} - x_\star^{r_t+1}\| \leqslant \lim_{t \to \infty} \gamma \|x^{r_t+1} - x^{r_t}\| = 0 \ \ \& \ \ \lim_{t \to \infty} \|x^{r_t} - x_\star^{r_t}\| \leqslant \lim_{t \to \infty} \gamma \|x^{r_t} - x^{r_t-1}\| = 0 \tag{42}$$

Thus, $\lim_{t \to \infty} e^{r_t} = \lim_{t \to \infty} e^{r_t+1} = 0$, which means

$$\bar{u} = \lim_{t \to \infty} u^{r_t} = -\lim_{t \to \infty} \left(\nabla f(x^{r_t}) - e^{r_t}\right) = -\nabla f(\bar{x}) \tag{43}$$

$$\lim_{t \to \infty} u^{r_t+1} = -\lim_{t \to \infty} \left(\nabla f(x^{r_t+1}) - e^{r_t+1}\right) = -\nabla f(\bar{x}) \tag{44}$$

Thus, $\lim_{t \to \infty} u^{r_t+1} = \bar{u}$.

Also, as $\mathcal{S}$ is finite, there exists a large enough T, such that $z^{r_t} = \bar{y}$ for $t \geqslant T$. Again due to the fact that $\mathcal{S}$ is finite, we can re-fine the sub-sequence such that $z^{r_t+1} = \hat{y}$. Thus, without loss of generality assume that these two conditions hold, i.e. $z^{r_t} = \bar{y}$ and $z^{r_t+1} = \hat{y}$ for all $t$ for an appropriately refined sub-sequence. This means that

$$\hat{y} \in \arg\min_x \|x - (x^{r_t} + \lambda^{-1} u^{r_t})\| \tag{45}$$

Moreover, $u^{r_t+1} = u^{r_t} + \lambda(x^{r_t+1} - \hat{y})$. Taking the $\lim_{t \to \infty}$ from both sides, we get

$$\hat{y} = \bar{x}. \tag{46}$$

Combining the above with Equation 45 we can easily see that

$$\|\bar{x} - (x^{r_t} + \lambda^{-1} u^{r_t})\| \leqslant \|a_i - (x^{r_t} + \lambda^{-1} u^{r_t})\|, \ i = 0, \cdots, N \tag{47}$$

Taking the limits $\lim_{t \to \infty}$ from both hand sides of the inequality for all the points $a_i$ we have

$$\|\bar{x} - (\bar{x} + \lambda^{-1}\bar{u})\| \leqslant \|a_i - (\bar{x} + \lambda^{-1}\bar{u})\|, \ i = 0, \cdots, N. \tag{48}$$

Thus,

$$\bar{x} \in \arg\min_{x \in \mathcal{S}} \|x - (\bar{x} - \lambda^{-1}\nabla f(\bar{x}))\|, \tag{49}$$

where we used the fact that $\bar{u} = -\nabla f(\bar{x})$. $\qquad \square$

Table 4: Global hyperparameters of ELSA shared across all models.

| Hyperparameter | Value |
|---|---|
| LR schedule | Linear decay |
| Interval $k$ | 32 |
| Adam $(\beta_1, \beta_2)$ | (0.9, 0.999) |
| Total # datapoints | 32768 |

Table 5: Learning rate ($\eta$), penalty ($\lambda$), and penalty schedule configuration across models at different sparsity levels.

| | Sparsity | $\lambda$ sched. | OPT-125M | OPT-1.3B | Gemma-2-2B | LLaMA-3.2-3B | LLaMA-2-7B | LLaMA-2-13B |
|---|---|---|---|---|---|---|---|---|
| $\eta$ | 50% | – | 1e-4 | 1e-4 | | 5e-5 | 5e-5 | |
| | 60% | – | 2e-4 | | 2e-5 | 1e-4 | 1e-4 | |
| | 70% | – | 1e-4 | 5e-5 | | 5e-5 | 5e-5 | 5e-5 |
| | 80% | – | 2e-4 | 1e-4 | | 1e-4 | | |
| | 90% | – | | | 5e-5 | | 1e-4 | |
| $\lambda$ | 50% | constant | 1e-4 | 1e-3 | 2e-1 | 1e-3 | 1e-3 | 2e-3 |
| | 60% | constant | 1e-3 | | | 2e-3 | 5e-3 | 1e-3 |
| | 70% | cosine | 2e-3 | 5e-3 | 1e-2 | 5e-3 | 2e-2 | 2e-2 |
| | 80% | cosine | 1e-3 | 1e-3 | 5e-3 | 1e-3 | | 5e-2 |
| | 90% | cosine | | | 5e-4 | | 1e-3 | 2e-3 |

Table 6: Batch size (BS) and number of steps used at different sparsity levels.

| Model(s) | 50–60% | | 70–90% | |
|---|---|---|---|---|
| | BS | Steps | BS | Steps |
| OPT-125M | 16 | 2048 | 8 | 4096 |
| OPT-1.3B, Gemma-2-2B, LLaMA-3.2-3B, LLaMA-2-7B, LLaMA-2-13B | 32 | 1024 | 8 | 4096 |

# B EXPERIMENTAL DETAILS

## B.1 IMPLEMENTATION AND REPRODUCTION DETAILS

Our implementation is based on PyTorch (Paszke et al., 2019), using the HuggingFace `transformers` and `datasets` libraries for model and data loading. ELSA is implemented over HuggingFace `Trainer`, supporting distributed training via PyTorch FSDP-2 (Zhao et al., 2023) with HuggingFace `Accelerate`.

All experimental results in this work are obtained with unified codebase, while baseline methods are reproduced using their original implementations whenever available. The environment configuration (dependencies, versions, and training scripts) can be found at `https://github.com/log-postech/elsa`.

Experiments are conducted on NVIDIA A100/H200 GPUs, with the number of GPUs scaled to model size: 2×GPUs for 1.3B–3B models, 4×A100 GPUs for 7B models, and 4×H200 GPUs for 13B and 27B models.

## B.2 DETAILS FOR SECTION 5.1

**Calibration/Training data.** To obtain baseline results (Wanda, SparseGPT, ALPS, L-ADMM, SAFE, SparseLLM), we follow the convention of Frantar & Alistarh (2023), sampling 128 calibration sequences from the C4 dataset with sequence length 2048. For ELSA, we adopt the same strategy, but use larger calibration sets to account for the iterative nature of our optimization.

**Training details.** We train ELSA with 32,768 data points, where each data point has sequence length of 2048. Batch size and number of steps differ across model, which is presented at table Table 6. We use Adam as the base optimizer. The penalty parameter is kept constant for moderate sparsity levels (50-60%), while we use cosine schedule, which gradually increases the penalty parameter from

0 at the start to $\lambda$ at the end of training. All model parameters and optimizer states uses full precision for training (except for memory-efficient setting and ablations), and automatic mixed precision with bf16 precision is used for efficient training. A full list of hyperparameter configurations is provided in Tables 4 and 5.

**Evaluation.** Perplexity is measured on the held-out (validation) C4 (Raffel et al., 2020) and Wikitext2 (Merity et al., 2017) datasets. Zero-shot performance is evaluated with `lm-eval-harness` across seven standard tasks: ARC-Easy/Challenge (ARC-E/C) (Clark et al., 2018), BoolQ (Clark et al., 2019), HellaSwag (Zellers et al., 2019), OpenBookQA (OBQA) (Mihaylov et al., 2018), RTE (Zeng & Urtasun, 2018), and Winogrande (Sakaguchi et al., 2021), and we report the average accuracy as in Section 5.1.

### B.3 DETAILS FOR SECTION 5.2

For the baseline (Wanda+LoRA / Full), we first prune the pretrained model with Wanda, and then retrain the remaining parameters using either LoRA (Hu et al., 2022) or full fine-tuning (Full). For a fair comparison, we use the same number data points with same sequence length as ELSA, with hyperparameters tuned separately for each method.

### B.4 DETAILS FOR SECTION 5.4

We ran ELSA$_\text{L}$ on Gemma-2-27B using 4×H200 GPUs. Fp8 representations for ADMM states $(u, z)$ were implemented based on the `torchao` framework(torchao, 2024), where we further extended the implementation to fully support `DTensor`, as required by the FSDP-2 framework for distributed training. For this setting, we used a learning rate of $\eta = 2 \times 10^{-5}$ and penalty parameter $\lambda = 0.002$, using cosine penalty scheduling.

### B.5 DETAILS FOR APPENDIX C.1

For N:M semi-structured sparsity, we use the same hyperparameter configuration as for 50% unstructured sparsity.

For non-uniform sparsity comparisons, we evaluate ELSA on LLaMA-3-8B using the hyperparameters of LLaMA-2-7B at 70% sparsity, while the results of SparseGPT, OWL, and EvoPress are taken directly from Sieberling et al. (2024). For ELSA (EvoPress), we adopt the non-uniform sparsity configurations provided in the official EvoPress repository, and initialize ELSA with these sparsity budgets while keeping the same training hyperparameters.

### B.6 DETAILS FOR APPENDIX C.2

For objective ablation, we used the OPT-125M model at 90% sparsity, fixing the total number of optimization steps to 4,096 and varying the data count from 256 up to 32,684, using the same hyperparameter configurations as in Table 5.

## C ADDITIONAL RESULTS

Here we provide additional results for different sparsity patterns (Appendix C.1), ablation studies on objective function and projection step (Appendix C.2), and a numerical results used to make visual plots in the main text followed by additional result reporting LLaMA-2-13B zero-shot task accuracy.

Table 8: Perplexity and zero-shot prediction accuracy of LLaMA-2-7B under N:M semi-structured sparsity. ELSA compares competitively to other methods, demonstrating its adaptivity. Note that 2:4 and 4:8 patterns are only 50% sparsity levels.

| Sparsity | Method | Perplexity (↓) | | Tasks (↑) | | | | | | | |
|---|---|---|---|---|---|---|---|---|---|---|---|
| | | Wiki | C4 | ARC-C | ARC-E | BoolQ | HellaSwag | OBQA | RTE | Winogrande | Avg. |
| 0% | Dense | 5.47 | 7.26 | 43.35 | 76.26 | 77.68 | 57.14 | 31.40 | 62.82 | 69.06 | 59.67 |
| 2:4 | Magnitude | 37.76 | 74.66 | 30.12 | 61.87 | 59.85 | 45.45 | 21.80 | 52.35 | 61.01 | 47.49 |
| | Wanda | 12.13 | 15.63 | 30.46 | 61.83 | 68.26 | 41.28 | 24.20 | 53.07 | 62.51 | 48.80 |
| | SparseGPT | 10.87 | 13.61 | 30.97 | 64.06 | 67.61 | 43.47 | 24.20 | 56.32 | 66.38 | 50.43 |
| | L-ADMM | 10.19 | 12.51 | 32.85 | 66.04 | 68.81 | 45.05 | 25.40 | 56.32 | 66.38 | 51.55 |
| | ALPS | 9.945 | 12.09 | 34.47 | 68.86 | 73.79 | 49.40 | 27.60 | 55.60 | 67.25 | 53.85 |
| | SAFE | 9.914 | 12.53 | 30.46 | 63.43 | 66.42 | 44.66 | 21.60 | 53.07 | 61.80 | 48.78 |
| | SparseLLM | 11.29 | 13.95 | 30.55 | 61.91 | 71.10 | 43.62 | 24.40 | 57.40 | 65.82 | 50.69 |
| | ELSA | 10.15 | 12.34 | 31.49 | 61.24 | 66.36 | 47.87 | 23.60 | 52.71 | 63.85 | 49.59 |
| 4:8 | Magnitude | 15.91 | 31.60 | 36.01 | 64.81 | 63.09 | 50.05 | 26.00 | 52.35 | 62.19 | 50.64 |
| | Wanda | 8.603 | 11.33 | 34.47 | 67.05 | 72.87 | 46.98 | 26.80 | 54.15 | 66.93 | 52.75 |
| | SparseGPT | 8.508 | 10.81 | 34.81 | 68.56 | 71.77 | 48.26 | 27.80 | 56.68 | 68.11 | 53.71 |
| | L-ADMM | 8.12 | 10.37 | 35.58 | 68.18 | 72.48 | 49.45 | 28.80 | 58.12 | 67.17 | 54.25 |
| | ALPS | 8.103 | 10.29 | 33.28 | 65.19 | 68.75 | 45.96 | 26.20 | 55.96 | 65.98 | 51.62 |
| | SAFE | 8.043 | 10.47 | 31.57 | 66.84 | 68.04 | 48.55 | 23.40 | 53.07 | 65.04 | 50.93 |
| | SparseLLM | 8.679 | 11.04 | 34.90 | 68.35 | 75.14 | 48.28 | 26.20 | 56.68 | 66.46 | 53.71 |
| | ELSA | 9.20 | 11.47 | 32.25 | 64.69 | 69.42 | 49.90 | 27.40 | 53.07 | 63.22 | 51.42 |

## C.1 OTHER SPARSITY PATTERNS

In this section, we analyze whether ELSA can adapt to other sparsity patterns including (i) N:M semi-structured sparsity and (ii) non-uniform sparsity over different layers.

We first evaluate ELSA for its adaptivity to N:M semi-structured sparsity, a setting designed for some current hardwares to accelerate computations (Sun et al., 2024; Fang et al., 2024). The results of both perplexity and zero-shot prediction accuracy are reported in Table 8. ELSA is roughly on par with existing methods, and yet, it is noteworthy that these 2:4 and 4:8 sparsity patterns only ensure 50% sparsity. More importantly, these results indicate that ELSA can easily adapt to arbitrary constraints of moderate sparsity levels without much trouble.

Table 7: Perplexity of LLaMA-3-8B at 70% sparsity. ELSA outperforms prior allocation methods.

| Method | Wiki(↓) | C4(↓) |
|---|---|---|
| SparseGPT | 85.84 | 98.35 |
| OWL | 48.07 | 52.32 |
| EvoPress | 28.76 | 33.72 |
| ELSA (EvoPress) | 26.11 | 29.33 |
| ELSA | 24.97 | 29.09 |

We also compare ELSA with non-uniform sparsity allocation based pruning methods. Specifically, we compare to OWL (Yin et al., 2024a) that allocates sparsity based on outlier distributions and to EvoPress (Sieberling et al., 2024) that uses an evolutionary search strategy to determine the non-uniform sparsity levels over different layers. We further set up a method that overrides ELSA with the mask found by the evolutionary strategy of EvoPress. Note that the sparsity level is set to be 70%; it is simply because these methods only works or reports up to this level. The results are presented in Table 7. One can see that ELSA substantially outperforms OWL and shows an improvement over EvoPress as well: to elaborate, for instance, it achieves the C4 perplexity of 29.09, compared to 33.72 for EvoPress and 52.32 for OWL. Notably, adopting the non-uniform mask found by EvoPress within ELSA yields some gains over the EvoPress itself, but it still falls short of the uniform allocation in ELSA, demonstrating the strength of our surrogate-free global formulation.

## C.2 ABLATIONS

In this section, we present two ablation analyses on (i) the choice of objective comparing the next token prediction (NTP) against the reconstruction error minimization (REM), and (ii) the projection step contrasting our objective-aware variant with the standard projection method.

Specifically, we first set up a experiment where we measure how effectively our surrogate-free approach with NTP make use of data to preserve the original model performance while vayring the number of data samples. We compare that to the existing REM approach. The results are plotted in Figure 6. While REM tend to perform better than NTP at low data regime, but it soon starts to saturate as data counts increases producing diminishing returns. This is in stark constrat to NTP by which pruning performance keeps on improving quite drastically with more data. Notably, REM requires memory to store dense model predictions, which can grow prohibitively large as with large data. By contrast, NTP naturally benefits from additional data and continues to improve, enabling scalable LLM sparsity. This in part reveals the inherent limitation of surrogate objectives.

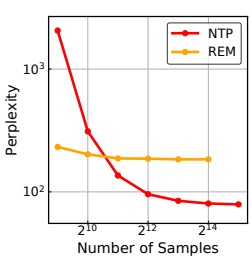

Figure 6: Effect of NTP on data efficiency and perplexity.

We also evaluate the effectiveness of the objective-aware projection on high-sparsity regimes. Specifically, we measure the perplexity of LLaMA-3.2-3B model pruned for 70-90% sparsity levels by turning on and off of the projection and report the results in Table 9. The benefit of objective-aware projection grows with sparsity: perplexity gap increases from 1.20 at 70% sparsity to 2.56 at 80%, and widens further at 90%. This demonstrates that incorporating objective-aware importance into the projection step can be beneficial particularly in high sparsity regimes.

Table 9: Effectiveness of geometric projection (✓).

| Sparsity | ✗ | ✓ |
|---|---|---|
| 70% | 29.44 | **28.24** |
| 80% | 40.06 | **37.50** |
| 90% | 65.41 | **48.69** |

## C.3 NUMERICAL RESULTS

Here we provide numerical results used to produce visual figures on the main text, followed by additional results reporting LLaMA-2-7B/13B zero-shot task accuracy.

Table 10: Perplexity (↓) of various models pruned with different methods across sparsity levels. Dense performance is shown under each model name (`Wiki` / `C4`). Results for SparseLLM on Gemma-2-2B and LLaMA-2-13B are omitted due to implementation limitations (e.g., architectural incompatibility, out-of-memory errors). We could not obtain results of SparseLLM in Gemma-2-2b, Llama-3.2-3B, Llama-2-13B.

| Model | Method | 50% Wiki | 50% C4 | 60% Wiki | 60% C4 | 70% Wiki | 70% C4 | 80% Wiki | 80% C4 | 90% Wiki | 90% C4 |
|---|---|---|---|---|---|---|---|---|---|---|---|
| **OPT-125M** (Dense: 27.65 / 26.56) | Magnitude | 193.4 | 141.0 | 920.0 | 598.2 | 3806 | 2263 | 4890 | 3213 | 6613 | 4475 |
| | Wanda | 38.93 | 34.91 | 77.85 | 63.33 | 351.8 | 248.9 | 1912 | 1066 | 4940 | 3126 |
| | SparseGPT | 37.02 | 33.51 | 60.90 | 49.83 | 239.2 | 156.3 | 2072 | 1050 | 6131 | 2443 |
| | L-ADMM | 33.02 | 31.21 | 45.04 | 38.49 | 100.5 | 74.61 | 580.8 | 315.8 | 3427 | 1350 |
| | ALPS | **32.70** | 30.91 | 43.07 | 36.94 | 90.85 | 66.28 | 484.8 | 267.7 | 2524 | 1094 |
| | SAFE | 33.88 | 30.54 | 47.21 | 37.46 | 120.1 | 75.2 | 1254 | 726.8 | 5382 | 2331 |
| | SparseLLM | 37.11 | 33.19 | 57.47 | 46.64 | 199.2 | 131.7 | 1576 | 752.2 | 4730 | 1825 |
| | ELSA | 37.68 | **30.10** | **41.5** | **33.22** | **49.57** | **39.86** | **65.30** | **47.74** | **95.33** | **62.28** |
| **OPT-1.3B** (Dense: 14.62 / 16.07) | Magnitude | 1712 | 403.3 | 9392 | 5066 | 9442 | 6498 | 1.6e4 | 1.1e4 | 2.9e4 | 1.8e4 |
| | Wanda | 18.42 | 20.62 | 26.82 | 28.77 | 105.7 | 94.98 | 2504 | 1181 | 1.3e4 | 8447 |
| | SparseGPT | 17.45 | 19.25 | 24.02 | 23.30 | 50.52 | 46.11 | 947.9 | 406.7 | 6472 | 2843 |
| | L-ADMM | 26.62 | 26.26 | 32.35 | 30.28 | 61.10 | 49.52 | 595.9 | 289.5 | 5659 | 2298 |
| | ALPS | 16.78 | 18.59 | 20.58 | 21.52 | 35.77 | 34.09 | 285.7 | 158.4 | 4590 | 1844 |
| | SAFE | **16.38** | **17.75** | **19.63** | 19.93 | 31.17 | 27.52 | 387.1 | 222.3 | 1.3e4 | 7544 |
| | SparseLLM | 17.73 | 19.40 | 23.23 | 24.03 | 56.36 | 47.96 | 861.7 | 372.0 | 5535 | 2217 |
| | ELSA | 18.99 | 18.45 | 22.11 | 20.2 | **27.13** | **24.43** | **36.89** | **31.51** | **61.52** | **45.39** |
| **Gemma-2-2B** (Dense: 8.71 / 13.16) | Magnitude | 51.66 | 57.68 | 2178 | 2064 | 4.4e7 | 3.5e6 | 2.5e9 | 2.4e8 | 5.0e9 | 2.3e9 |
| | Wanda | 12.07 | 17.49 | 21.39 | 32.40 | 117.5 | 152.0 | 994.6 | 855.6 | 1.1e4 | 5524 |
| | SparseGPT | 11.58 | 16.67 | 16.53 | 23.44 | 34.73 | 47.43 | 147.7 | 160.5 | 983.1 | 776.5 |
| | L-ADMM | 11.02 | 15.84 | 14.65 | 20.84 | 26.91 | 38.32 | 86.64 | 110.8 | 308.2 | 300.3 |
| | ALPS | **10.93** | **15.77** | **14.42** | 20.32 | 24.96 | 35.08 | 73.50 | 94.26 | 238.5 | 254.3 |
| | SAFE | 11.61 | 16.21 | 15.22 | 20.32 | 25.67 | 33.39 | 68.55 | 75.22 | 432.7 | 345.0 |
| | SparseLLM | — | — | — | — | — | — | —- | — | — | — |
| | ELSA | 13.05 | 17.22 | 15.93 | **19.83** | **21.22** | **24.55** | **30.29** | **31.68** | **49.37** | **44.93** |
| **LLaMA-3.2-3B** (Dense: 7.81 / 11.32) | Magnitude | 139.4 | 216.1 | 1.5e4 | 1.4e4 | 1.0e5 | 8.1e5 | 3.5e5 | 3.5e5 | 3.0e5 | 2.4e5 |
| | Wanda | 13.01 | 19.08 | 31.39 | 42.53 | 142.4 | 168.1 | 3859 | 1821 | 1.4e4 | 8766 |
| | SparseGPT | 12.27 | 17.41 | 23.38 | 30.47 | 86.88 | 84.12 | 292.9 | 237.1 | 1807 | 1094 |
| | L-ADMM | 11.56 | 16.32 | 19.06 | 24.84 | 45.48 | 53.30 | 160.4 | 126.9 | 760.5 | 509.5 |
| | ALPS | 11.31 | 15.88 | 18.16 | 22.83 | 41.79 | 46.48 | 166.32 | 109.0 | 542.0 | 367.0 |
| | SAFE | **10.68** | **15.51** | 16.76 | 22.57 | 50.78 | 57.86 | 330.9 | 267.2 | 3410 | 2343 |
| | SparseLLM | — | — | — | — | — | — | — | — | — | — |
| | ELSA | 12.06 | 17.34 | **16.65** | **21.73** | **24.07** | **28.24** | **36.25** | **37.50** | **50.88** | **48.69** |
| **LLaMA-2-7B** (Dense: 5.47 / 7.26) | Magnitude | 16.03 | 21.34 | 1924 | 2063 | 5.0e4 | 2.8e4 | NaN | NaN | NaN | NaN |
| | Wanda | 6.92 | 9.24 | 10.79 | 13.99 | 76.32 | 81.08 | 4096 | 2673 | 2.0e4 | 1.0e4 |
| | SparseGPT | 7.01 | 9.23 | 10.20 | 12.93 | 27.12 | 30.94 | 107.3 | 100.8 | 1430 | 864.5 |
| | L-ADMM | 6.80 | 8.97 | 9.40 | 11.47 | 20.56 | 22.20 | 60.78 | 58.63 | 400.5 | 287.1 |
| | ALPS | 6.86 | 9.02 | 9.33 | **11.30** | 19.39 | 20.37 | 48.43 | 47.22 | 248.8 | 180.9 |
| | SAFE | **6.72** | **8.87** | **9.02** | 11.40 | 86.80 | 48.54 | 8.1e5 | 5.3e5 | 1.6e4 | 1.6e4 |
| | SparseLLM | 7.23 | 9.51 | 10.74 | 13.25 | 37.65 | 35.00 | 126.5 | 94.28 | 1267 | 648.0 |
| | ELSA | 7.5 | 9.81 | 9.16 | 11.34 | **13.20** | **14.08** | **20.83** | **19.56** | **26.97** | **23.14** |
| **LLaMA-2-13B** (Dense: 4.88 / 6.73) | Magnitude | 6.83 | 9.38 | 11.82 | 14.62 | 214.2 | 191.9 | 3.9e4 | 4.9e4 | 7.5e4 | 6.5e4 |
| | Wanda | 5.97 | 8.30 | 8.40 | 11.53 | 45.37 | 56.27 | 1004 | 838.8 | 2.2e4 | 1.3e4 |
| | SparseGPT | 6.03 | 8.22 | 8.27 | 10.93 | 19.79 | 23.47 | 97.82 | 79.17 | 1442 | 984.1 |
| | L-ADMM | 5.92 | 8.11 | 7.57 | 10.05 | 14.81 | 17.56 | 44.78 | 44.42 | 391.1 | 242.1 |
| | ALPS | 5.90 | 7.99 | 7.56 | 9.92 | 14.17 | 16.28 | 38.44 | 36.78 | 231.3 | 152.1 |
| | SAFE | **5.73** | **7.82** | **6.90** | **9.24** | 12.47 | 14.57 | 93.49 | 73.25 | 2122 | 1388 |
| | SparseLLM | — | — | — | — | — | — | — | — | — | — |
| | ELSA | 6.54 | 8.78 | 7.96 | 9.93 | **11.14** | **12.20** | **17.21** | **16.60** | **30.19** | **25.07** |

Table 11: Zero-shot accuracy (%) of Llama-2-7B across multiple tasks, in various sparsity regime (50%-90%).

| Sparsity | Method | Tasks | | | | | | | |
|---|---|---|---|---|---|---|---|---|---|
| | | ARC-C | ARC-E | BoolQ | HellaSwag | OBQA | RTE | Winogrande | Avg |
| 0% | Dense | 43.35 | 76.26 | 77.68 | 57.14 | 31.40 | 62.82 | 69.06 | 59.67 |
| 50% | Magnitude | 34.90 | 64.02 | 62.91 | 49.13 | 26.80 | 57.04 | 63.22 | 51.14 |
| | Wanda | 39.25 | 72.22 | 75.17 | 52.64 | 30.60 | 53.43 | 67.17 | 55.78 |
| | SparseGPT | 38.23 | 71.34 | 75.99 | 52.70 | 29.80 | 56.32 | **69.77** | 56.31 |
| | L-ADMM | 39.68 | 72.77 | **76.24** | 53.35 | **31.40** | **61.37** | 69.30 | **57.73** |
| | ALPS | **40.61** | **72.90** | 75.44 | **53.37** | 30.80 | 57.76 | 68.98 | 57.14 |
| | SAFE | 38.14 | 72.14 | 74.83 | 52.15 | 26.00 | 57.04 | 66.77 | 55.30 |
| | SparseLLM | 38.05 | 71.25 | 75.14 | 52.66 | 29.60 | 53.43 | 69.30 | 55.63 |
| | ELSA | 39.42 | 71.30 | 73.03 | 53.12 | 29.40 | 58.48 | 66.54 | 56.39 |
| 60% | Magnitude | 25.17 | 44.87 | 47.80 | 35.00 | 20.00 | 50.90 | 53.12 | 39.55 |
| | Wanda | 30.63 | 64.44 | 65.51 | 43.51 | 25.80 | 54.15 | 64.01 | 49.72 |
| | SparseGPT | 31.57 | 64.06 | **72.57** | 45.0 | 25.80 | 53.43 | 65.51 | 51.13 |
| | L-ADMM | 34.13 | **66.50** | 70.43 | 47.29 | 26.60 | **55.60** | **66.61** | **52.45** |
| | ALPS | **34.38** | 66.33 | 70.64 | 47.81 | **27.2** | 54.15 | 66.29 | 52.40 |
| | SAFE | 31.14 | 64.14 | 71.10 | 46.43 | 24.00 | 54.15 | 62.98 | 50.57 |
| | SparseLLM | 32.59 | 64.52 | 70.86 | 45.24 | 25.80 | 53.79 | 66.14 | 51.28 |
| | ELSA | 32.08 | 65.83 | 67.52 | **49.19** | 26.00 | 53.80 | 63.38 | 51.41 |
| 70% | Magnitude | 22.87 | 27.82 | 37.95 | 25.90 | 17.20 | 53.07 | 49.25 | 33.43 |
| | Wanda | 18.6 | 30.01 | 57.28 | 28.04 | 12.0 | 52.71 | 48.86 | 35.36 |
| | SparseGPT | 22.01 | 42.34 | **65.14** | 33.04 | 16.8 | 52.71 | 57.7 | 41.39 |
| | L-ADMM | 23.81 | 50.63 | 63.21 | 36.57 | 20.40 | 54.15 | 60.77 | 44.22 |
| | ALPS | 25.51 | 52.78 | 63.46 | 37.54 | 20.8 | 53.43 | 61.72 | 45.03 |
| | SAFE | 24.23 | 45.62 | 43.76 | 34.74 | 18.40 | 52.71 | 53.12 | 38.94 |
| | SparseLLM | 20.90 | 40.32 | 61.87 | 32.74 | 16.0 | **54.51** | 57.46 | 40.54 |
| | ELSA | **27.13** | **55.81** | 63.61 | **43.16** | **22.40** | 52.71 | 58.64 | **46.21** |
| 80% | Magnitude | **22.35** | 25.38 | 43.67 | 25.72 | 13.00 | 46.57 | 51.62 | 32.62 |
| | Wanda | 20.82 | 26.98 | 37.83 | 25.89 | 15.0 | 52.71 | 49.25 | 32.64 |
| | SparseGPT | 17.92 | 27.95 | 38.07 | 27.51 | 12.0 | **53.07** | 49.01 | 32.22 |
| | L-ADMM | 18.26 | 29.29 | 57.49 | 28.33 | 13.00 | **53.07** | 51.22 | 35.81 |
| | ALPS | 19.37 | 32.07 | **61.1** | 29.06 | 12.6 | 52.71 | 50.91 | 36.83 |
| | SAFE | 21.76 | 25.80 | 37.83 | 26.01 | 14.00 | 52.71 | 49.80 | 32.56 |
| | SparseLLM | 18.09 | 28.70 | 43.55 | 27.57 | 11.6 | 52.71 | 48.86 | 33.01 |
| | ELSA | 20.99 | **44.61** | 60.67 | **34.02** | **16.80** | 52.71 | **53.20** | **40.43** |
| 90% | Magnitude | **22.78** | 25.93 | 39.17 | 25.53 | 16.0 | 47.29 | 50.12 | 32.40 |
| | Wanda | 21.67 | 25.46 | 37.83 | 25.83 | 15.2 | 47.29 | 49.33 | 31.8 |
| | SparseGPT | 20.65 | 26.77 | 37.83 | 25.7 | 13.0 | 52.71 | 50.59 | 32.46 |
| | L-ADMM | 19.97 | 26.14 | 37.83 | 26.46 | 13.60 | 51.62 | 47.51 | 31.88 |
| | ALPS | 19.45 | 26.89 | 37.8 | 26.81 | 12.8 | **53.79** | 46.65 | 32.03 |
| | SAFE | 21.84 | 26.52 | 37.83 | 25.91 | 15.80 | 52.71 | 47.83 | 32.63 |
| | SparseLLM | 20.56 | 25.72 | 37.83 | 25.94 | 13.8 | 52.71 | 46.96 | 31.93 |
| | ELSA | 18.52 | **41.33** | **57.25** | **31.54** | **16.60** | 52.71 | **51.70** | **38.52** |

Table 12: Zero-shot accuracy (%) of Llama-2 13B across multiple tasks, under various sparsity levels. We could not obtain SparseLLM in Llama-2-13B.

| Sparsity | Method | Tasks | | | | | | | |
|---|---|---|---|---|---|---|---|---|---|
| | | ARC-C | ARC-E | BoolQ | HellaSwag | OBQA | RTE | Winogrande | Avg |
| 0% | Dense | 48.46 | 79.38 | 80.55 | 60.04 | 35.20 | 65.34 | 72.14 | 63.02 |
| 50% | Magnitude | 38.48 | 70.58 | 57.65 | 54.39 | 27.80 | 55.96 | 65.35 | 52.89 |
| | Wanda | 43.09 | **76.30** | 80.95 | 56.96 | 31.20 | 60.65 | 71.43 | 60.08 |
| | SparseGPT | 42.41 | 74.96 | **81.53** | 55.95 | 31.00 | **64.26** | 71.35 | 60.21 |
| | L-ADMM | 43.17 | 75.84 | **82.29** | 56.51 | 32.00 | 63.18 | 71.98 | **60.71** |
| | ALPS | 42.66 | **76.30** | 81.22 | 56.71 | 32.60 | 62.82 | **72.14** | 60.64 |
| | SAFE | 41.64 | 75.84 | 80.40 | 56.59 | 30.60 | 60.65 | 69.14 | 59.27 |
| | ELSA | **43.68** | 75.12 | 77.71 | **57.45** | **33.80** | 55.60 | 70.00 | 59.05 |
| 60% | Magnitude | 27.13 | 56.14 | 47.49 | 44.66 | 21.80 | 52.71 | 57.46 | 43.93 |
| | Wanda | 37.97 | 68.81 | 77.16 | 48.71 | 28.20 | 59.57 | 68.19 | 55.51 |
| | SparseGPT | 36.01 | 69.40 | 78.72 | 49.38 | 27.4 | 57.76 | 70.56 | 55.60 |
| | L-ADMM | 39.76 | **72.98** | 80.70 | 51.49 | 29.8 | 59.93 | 70.32 | 57.86 |
| | ALPS | 40.44 | 72.93 | **81.68** | 51.97 | 30.80 | **60.29** | **71.90** | **58.58** |
| | SAFE | 36.95 | 72.43 | 78.38 | 52.09 | 28.80 | 57.40 | 67.88 | 56.28 |
| | ELSA | **40.53** | 70.67 | 75.17 | **54.22** | **31.00** | 53.43 | 67.80 | 56.12 |
| 70% | Magnitude | 20.65 | 31.31 | 38.65 | 27.53 | 14.60 | 52.71 | 49.25 | 33.53 |
| | Wanda | 18.43 | 36.45 | 62.35 | 29.25 | 13.0 | 52.71 | 50.83 | 37.57 |
| | SparseGPT | 25.34 | 49.58 | 67.86 | 36.27 | 20.2 | 52.71 | 60.93 | 44.70 |
| | L-ADMM | 27.56 | 59.64 | 69.76 | 40.05 | 24.00 | **53.43** | 65.35 | 48.54 |
| | ALPS | 29.61 | 61.20 | 70.09 | 40.86 | **26.6** | 53.07 | 64.56 | 49.43 |
| | SAFE | 29.78 | 61.07 | 69.17 | 41.62 | 20.20 | 52.71 | 58.96 | 47.64 |
| | ELSA | **34.13** | **62.42** | **70.12** | **47.52** | 24.80 | 52.71 | 60.38 | **50.30** |
| 80% | Magnitude | 21.84 | 25.63 | 41.80 | 25.88 | 14.80 | **53.07** | 49.25 | 33.18 |
| | Wanda | 20.48 | 26.26 | 37.83 | 26.81 | 12.6 | 52.71 | 50.04 | 32.39 |
| | SparseGPT | 19.62 | 28.79 | 59.05 | 27.77 | 12.8 | 52.71 | 49.33 | 35.72 |
| | L-ADMM | 19.11 | 33.80 | 62.14 | 29.67 | 14.60 | 52.71 | 53.28 | 37.90 |
| | ALPS | 20.05 | 35.99 | 62.17 | 30.65 | 14.0 | 52.71 | **54.93** | 38.64 |
| | SAFE | 18.34 | 28.37 | 40.64 | 27.44 | 12.80 | 52.71 | 50.51 | 32.97 |
| | ELSA | **24.32** | **50.97** | **63.52** | **38.03** | **19.60** | 52.71 | 53.75 | **43.27** |
| 90% | Magnitude | 21.42 | 24.87 | 44.16 | 25.72 | 15.0 | 46.57 | 51.78 | 32.79 |
| | Wanda | 21.33 | 25.93 | 37.83 | 25.80 | 13.8 | 52.71 | 51.54 | 32.70 |
| | SparseGPT | 21.08 | 25.76 | **58.62** | 25.87 | 13.8 | 52.35 | 49.49 | 35.28 |
| | L-ADMM | 19.88 | 26.01 | 39.45 | 27.08 | 13.80 | **53.79** | 50.04 | 32.86 |
| | ALPS | 18.94 | 26.94 | 43.52 | 27.37 | 13.4 | 52.71 | 48.30 | 33.02 |
| | SAFE | **22.10** | 25.76 | 37.83 | 26.02 | 14.20 | 52.71 | **53.43** | 33.15 |
| | ELSA | 19.03 | **36.15** | 58.44 | **28.65** | **16.20** | 52.71 | 50.43 | **37.37** |

