# OpenReview forum: "The Unseen Frontier: Pushing the Limits of LLM Sparsity with Surrogate-Free ADMM"
_ICLR.cc/2026/Conference — ICLR 2026 Poster_

### Official Review · Reviewer_oqwk · 2025-10-30

**Soundness:** 3
**Presentation:** 3
**Contribution:** 2
**Rating:** 6
**Confidence:** 4

**Summary:**

This paper addresses the "sparsity wall" (50--60\%) in large language model (LLM) pruning by introducing ELSA (Extreme LLM Sparsity via Surrogate-Free ADMM). ELSA formulates pruning as minimizing the true task loss $f(x)$ under an $\ell_0$ sparsity constraint and solves it via ADMM, which alternates between weight optimization and projection onto the sparse set. Instead of layer-wise reconstruction surrogates, it performs an objective-aware projection using diagonal Fisher/Adam second-moment statistics to guide weighted top-$k$ selection. Experiments on OPT, Gemma-2, and LLaMA-2 models show stable perplexity up to 90\% sparsity and consistent gains over SparseGPT, Wanda, ALPS, L-ADMM, and SAFE. The paper also mentions a quantized extension (ELSA-L) for 27B-scale models with theoretical guarantees, though detailed results appear later in the text.

**Strengths:**

1. The curvature-weighted projection is simple yet effective, using readily available second-moment statistics to guide pruning decisions.

2. Empirical results are strong and consistent, showing stable perplexity up to 90% sparsity across diverse LLM architectures.

3. The paper is well written and conceptually cohesive. The additional theories round up a solid work.

**Weaknesses:**

1. The experiments convincingly show perplexity improvements, but omit practical metrics such as wall-clock time, memory footprint, and actual inference acceleration. In reality, pruning methods—whether extremely local (e.g., SparseGPT, Wanda) or global (e.g., this work, SparseLLM and etc)—ultimately represent different points on a performance–cost tradeoff curve. With sufficient GPU resources, even global pruning becomes computationally feasible, reducing the motivation for additional algorithmic complexity like ADMM. It would strengthen the paper to explicitly highlight the practical scenarios or deployment constraints where ELSA provides tangible benefits over simpler baselines.

2. The evaluation focuses solely on perplexity and task accuracy, without measuring actual inference acceleration, FLOPs reduction, or memory throughput gains after pruning. This limits the practical impact, as sparsity alone does not guarantee real-world speed-ups.

3. While the paper frames pruning as a constrained optimization problem and leverages ADMM elegantly, both the global-pruning perspective and ADMM-style decomposition have precedents (e.g., SparseLLM, L-ADMM, SAFE). The novelty lies mainly in unifying these ideas under a “surrogate-free” formulation rather than introducing a fundamentally new optimization principle.

4. [minor] Despite claiming global sparsity, the projection step is performed per-tensor using diagonal curvature estimates, so cross-layer dependencies are not modeled. The approach therefore remains an efficient approximation to the ideal global objective rather than a full solution.

**Questions:**

1. How does ELSA translate its high sparsity into real inference acceleration or memory savings on modern hardware?

2. Can you quantify the additional compute cost (training or calibration time) introduced by ADMM compared to simpler methods like SparseGPT?

3. How robust is the whole approach to the hyper-parameters? ADMM-based approaches performance could be quite sensitive to hyper-parameters and hard to tune in practice, especially for large-scale LLM pruning problem IMHO.

---

> ### Author Response · Authors · 2025-11-25
> **Response to reviewer oqwk (1/4)**
>
> We sincerely appreciate the reviewer’s thoughtful reading and generous comments. Your recognition of the *curvature-weighted projection, empirical performance across sparsity levels* was truly encouraging. Your feedback helped us further clarify and strengthen the depth of our paper. We address each of your points in detail below and would be glad to continue the discussion.
>
>
> ---
> **W1: Practical scenario or deployment constraints of ELSA’s benefits**
>
> > ``It would strengthen the paper to explicitly highlight the practical scenarios or deployment constraints where ELSA provides tangible benefits over simpler baselines.’’
>
> We sincerely appreciate the reviewer’s constructive suggestion. `ELSA` does indeed offer tangible benefits, most notably in scenarios where large models must be served persistently. LLM-based service providers are one representative example, where reducing the overall cost primarily requires targeting the much more dominant, long-term workload of inference serving, rather than the pruning stage, which is a comparatively one-time procedure (i.e., computational savings occur at deployment). Achieving high sparsity would be a key to enabling this, something that prior layerwise methods have effectively traded away in pursuit of reduced pruning cost, whereas `ELSA` excel and, thus would provide effective inference acceleration and memory savings in sparse model serving and deployment environments [Hoefler+21].
>
> Doing so necessitates the use of specialized sparse inference engines, which is fortunately a highly active line of research [Gale+20, Pool+21, Zheng+22, Xia+23, N+23, Hu+24, Fan+25]. `ELSA`’s ability to reach extreme sparsity allows it to leverage these systems more effectively, translating its sparsity into real deployment gains. More thorough discussion—including relevant metrics such as inference acceleration, memory savings (W2 & Q1), wall-clock behavior, and additional cost (Q2)—is provided in subsequent responses.
>
> Once again, we thank the reviewer for the valuable feedback and will incorporate the above discussion into the revised Discussion section.
>
> ---
>
> **W2&Q1: inference acceleration and memory savings from high sparsity**
>
> > “How does ELSA translate its high sparsity into real inference acceleration or memory savings on modern hardware?”
>
> It is possible to gain real inference acceleration or memory savings at any sparsity, and especially at high-sparsity, and have long been recognized and remain an active research area today [Gale+20, Zheng+22, N+23, Xia+23, Fan+25, Joo+25].
>
> **Inference speedup**: [Xia+23] reports that for OPT-13B/30B/175B, 70% sparsity yields on average x1.4 speedup, and 90% sparsity yields up to x2.1 speedup. This has further potential when various well-designed sparsity patterns are introduced [Zhou+25, Liu+22, Kang+19], notable example being 2:4 semi-structured pattern and its native support by NVIDIA GPUs to enable up to ×2 faster inference [Pool+21, Hu+24]. We are currently conducting experiments to quantify the achievable acceleration in our setting, and will update the paper with the results once they are finalized.
>
> **Memory savings**: Memory savings can also be achieved by storing sparse matrices in various dedicated formats such as COO and CSR [Hoefler+21]. CSR format [N+23], in particular, is much more effective with higher sparsity, reaching up to x3 memory savings for storing LLaMA-2-7B at 90% sparsity as opposed to 50% sparsity requiring additional 1.4 GB of storage due to indexing overheads. Having parameters be half precision further improves the compression ratio to ×6.14.
>
> | metric | dense | 50% sparse | 90% sparse | 90% sparse + fp16 |
> |-|-|-|-|-|
> | memory usage  | 25.70 GB   | 27.10 GB (×0.94) | 8.36 GB (×3.07)  | 4.18 GB (×6.14)   |
>
> Thus, both inference acceleration or memory savings are expected to benefit from extreme sparsity of `ELSA`, which therefore positions it as advancing the practical utility of sparsity for inference efficiency. We will incorporate this discussion into the final version of the paper.

---

> ### Author Response · Authors · 2025-11-25
> **Response to reviewer oqwk (2/4)**
>
> **W3: Novelty**
>
> > ``The novelty lies mainly in unifying these ideas under a “surrogate-free” formulation rather than introducing a fundamentally new optimization principle.’’
>
> `ELSA` differs from existing precedents along several key dimensions with the singular goal of enhancing its effectiveness under the surrogate-free objective. Precisely, the proposed objective-aware projection—which leverages information directly from the surrogate-free/global objective—and the quantized states introduced for scalability, represent clear distinctions from existing ADMM-based pruning algorithms. Notably, the convergence guarantees of prior methods do not directly apply to this new formulation, necessitating a novel convergence analysis (Section 3.3), which we believe supports its difference among prior approaches. A more in-depth discussion of these differences can be found at our response to reviewer `KMnE`(W1).
>
> It is also worth emphasizing that many influential works now positioned as central in the machine learning community have been recognized for introducing innovations that, while being seemingly modest relative to being ``fundamentally new'', still has led to substantial empirical improvements. For example, ResNet [He+16] differs from prior CNN architectures primarily by the addition of simple skip connections, yet it delivered dramatic performance improvements and is now the most cited papers of the 21st century. Similarly, AdamW, now the de-facto optimizer for training Transformers, differs algorithmically from Adam only in the decoupling of weight decay, but this minor structural change leads to significantly improved performance and has effectively replaced Adam in modern practice. In the same spirit, `ELSA` introduces sufficient differences that have collectively yielded substantial performance improvements, and we believe these insights could facilitate a paradigm shift in the way LLM pruning is approached.
>
> We appreciate the reviewer’s thoughtful comment, and we will revise the manuscript to ensure these points are clearly highlighted.
>
> ---
>
> **W4: Projection step**
>
> > ``[minor] Despite claiming global sparsity, the projection step is performed per-tensor using diagonal curvature estimates, so cross-layer dependencies are not modeled. The approach therefore remains an efficient approximation to the ideal global objective rather than a full solution.’’
>
> That is an excellent insight. While a fully global projection would be ideal, it is not feasible for LLMs in practice, so we instead adopt an efficient yet effective approximation. In the $z$-step, we exploit global gradient information of $f$, and in the $x$-step we directly optimize the surrogate-free/global objective, so that cross-layer dependencies are naturally captured in $x$. Because these procedures are unified within the ADMM framework, the cross-layer dependencies are indirectly passed to $z$ through the dual updates, so `ELSA`’s projection step also implicitly reflects them.
>
> Even with this approximate design, `ELSA` already shows strong empirical performance. That said, if a fully cross-layer–aware projection for LLMs were made tractable, it would likely bring further gains. We see this as a promising future direction and will aim to mention it in the discussion section. Thanks again for the thoughtful comment.

---

> ### Author Response · Authors · 2025-11-25
> **Response to reviewer oqwk (3/4)**
>
> **Q1: compute cost of pruners**
>
> > ``Can you quantify the additional compute cost (training or calibration time) introduced by ADMM compared to simpler methods like SparseGPT?’’
>
> Thank you for the thoughtful suggestion. In our experiments, we evaluated each method using its reported standard settings, and the resulting practical metrics and performance are summarized below (LLaMA-2-7B, A100-80GB, 90% sparsity):
>
> | method     | wall-clock time | # GPUs | Perf (wiki ppl) | Perf (C4 ppl) |
> |------------|-----------------|--------|------------------|----------------|
> | wanda      | 0.159 H         | 1      | 2.0e4            | 1.0e4          |
> | sparsegpt  | 0.251 H         | 1      | 1430             | 864.5          |
> | alps       | 12.57 H         | 1      | 248              | 180.9          |
> | elsa       | 1.78 H          | 4      | 26.97            | 23.14          |
>
> As shown in the table, ADMM-based methods (`ALPS`, `ELSA`) inevitably require more computation than simple one-shot approaches like `Wanda` and `SparseGPT`, due to their iterative nature, and thus naturally benefit from additional resources. However, even when other methods are allowed comparable or higher computational budgets, they still fail to reach `ELSA`’s performance in practice.
>
> For instance, `ALPS`, which is also ADMM-based and consumes more total compute than ELSA under our settings, still performs significantly worse. Likewise, giving `Wanda` the same compute budget as `ELSA` does not close the performance gap (please refer to our global response). These observations indicate that `ELSA` is not merely benefiting from additional compute, but instead genuinely improves the performance–cost tradeoff under high-resource regimes.
>
> We appreciate the reviewer’s comment and will revise the paper to ensure this point is clearly conveyed.
>
> ---
>
> **Q3: Hyperparameters sensitivity**
>
> > ``How robust is the whole approach to the hyper-parameters? ADMM-based approaches could be quite sensitive to hyper-parameters and hard to tune in practice, especially for large-scale LLM pruning, in my opinion.’’
>
> The primary training hyperparameter introduced by adopting ADMM is the penalty strength $\lambda$, which controls the sparsity constraint (there is also the interval $k$ for the $z$-update, but this parameter is not particularly sensitive in practice [Lee+25]). Based on our experience, careful tuning of $\lambda$ is indeed important for achieving strong results (see Table 5 in Appendix A). Nevertheless, the effective range of $\lambda$ is comparable to standard tuning knobs such as weight decay, and thus remains within a familiar and manageable scale.
>
> It is also worth noting that the need for hyperparameter tuning is unavoidable for any optimization-based pruning approach. Conversely, heuristic-based approaches often introduce additional hyperparameters of their own (i.e., gradual pruning schedule [Zhu & Gupta+17, Kurtic+23], lottery ticket hypothesis [Frankle+19]).
>
> We believe that using adaptive penalty update strategies [Boyd+11] could further reduce the tuning burden, and this represents a meaningful direction for future work. Last but not least, this tuning overhead applies only to the one-time pruning procedure; given that the pruned model is subsequently used for long-term serving, the relative cost of hyperparameter tuning is arguably quite minimal in the broader deployment context.

---

> ### Author Response · Authors · 2025-11-25
> **Response to reviewer oqwk (4/4)**
>
> **Remark**
>
> We sincerely hope that our detailed response has addressed all your comments thoroughly and satisfactorily. Should there be any remaining concerns or a need for further clarification, we would be most happy to provide additional details during the discussion period.
>
> We are greatly encouraged by your positive assessment of our work's core strengths. In particular, we appreciate your recognition of our method as a simple yet effective curvature-weighted projection utilizing second-moment statistics. Furthermore, we are pleased that you acknowledged our strong empirical results, showing stability up to 90% sparsity across diverse LLM architectures. Your comments on the paper’s superior conceptual cohesion and solid theoretical foundations also affirm the substantial contribution and rigorous nature of our work. Thank you again for your insightful review and dedicated time.
>
> ---
>
> **Reference**
>
> [Pool+21] Accelerating Inference with Sparsity Using the NVIDIA Ampere Architecture and NVIDIA TensorRT, Nvida blogpost \
> [Kurtic+23] Kurtic et al., Sparse Fine-Tuning for Inference Acceleration of Large Language Models, NeurIPS 2023 Workshop \
> [Boyd+11] Distributed optimization and statistical learning via the alternating direction method of multipliers, Foundations and Trends® in Machine learning.\
> [Frankle+19] The Lottery Ticket Hypothesis: Finding Sparse, Trainable Neural Networks, ICLR 2019 \
> [Fan+25] SpInfer: Leveraging Low-Level Sparsity for Efficient Large Language Model Inference on GPUs, EuroSys ‘25 \
> [Xia+23] Xia et al., Flash-LLM: Enabling Cost-Effective and Highly-Efficient Large
> Generative Model Inference with Unstructured Sparsity, VLDB 2023 \
> [N+23] NVIDIA. cuSPARSE Library. https://docs.nvidia.com/cuda/cusparse/index. 2023 \
> [Gale+20] Gale et al., Sparse gpu kernels for deep learning, SC20 2020 \
> [Zheng+22], Zheng et al., SparTA: Deep-Learning Model Sparsity via Tensor-with-Sparsity-Attribute, OSDI 2022 \
> [Hu+24] Hu et al., Accelerating Transformer Pre-training with 2:4 Sparsity, ICML 2024 \
> [Hoefler+21] Hoefler et al., Sparsity in Deep Learning: Pruning and growth for efficient inference and training in neural networks, JMLR 2021 \
> [Zhu & Gupta17] Zhu & Gupta, To prune, or not to prune: exploring the efficacy of pruning for model compression, arXiv \
> [Joo+25] Joo et al., Coruscant: Co-Designing GPU Kernel and Sparse Tensor Core to Advocate Unstructured Sparsity in Efficient LLM Inference, MICRO 2025 \
> [Zhou+25] Zhou et al., Learning N:M Fine-grained Structured Sparse Neural Networks From Scratch, ICLR 2021 \
> [Liu+22] Liu et al., S2TA: Exploiting Structured Sparsity for Energy-Efficient Mobile CNN Acceleration, HPCA 2022 \
> [Kang+19] Kang, Accelerator-Aware Pruning for Convolutional Neural Networks, IEEE 2019 \

---

> ### Author Response · Authors · 2025-11-26
> **Updated results regarding Q1**
>
> We have updated results regarding inference speedup. `ELSA` with 90% sparsity achieves a **2.5x** inference speedup with **4.6x** memory savings compared to the dense model (LLaMA-2-7b). Please see the global response for full details.

---

### Official Review · Reviewer_KMnE · 2025-10-31

**Soundness:** 4
**Presentation:** 3
**Contribution:** 3
**Rating:** 4
**Confidence:** 4

**Summary:**

This paper introduces ELSA, a surrogate-free ADMM-based method for pruning large language models (LLMs) to extreme sparsity levels (up to 90\%) while preserving performance. It critiques existing layer-wise reconstruction approaches for their limitations and compounding errors. ELSA directly optimizes the sparsity-constrained objective, achieving significantly lower perplexity (e.g., 7.8× better on LLaMA-2-7B at 90\% sparsity) and higher zero-shot accuracy across models like OPT, Gemma, and LLaMA.

**Strengths:**

1. This paper has a reasonable structure and is clearly written. The authors explain their method in detail and make substantial theoretical contributions.
2. The authors conducted extensive experiments across various model architectures and scales, evaluating metrics including perplexity and zero-shot task performance (e.g., ARC, BoolQ) and performing fair comparisons with state-of-the-art methods (e.g., SparseGPT, ALPS, SAFE). The results show that ELSA exhibits significant performance advantages at high sparsity levels (80-90\%).

**Weaknesses:**

1. Both ALPS and L-ADMM have also proposed ADMM-based pruning algorithms for LLMs. The authors should clearly elaborate on the differences between ELSA and ALPS to highlight the unique contributions of ELSA.
2. At low sparsity rates (50% and 60%), the accuracy of ELSA is significantly lower than that of the optimal L-ADMM baseline (Tables 7 and 8).
3. Although ELSA has advanced the performance of LLMs at high sparsity rates (70%-90%), there remains a substantial performance gap between ELSA and dense LLMs.
4. Pruning a 7B-parameter model requires 4 A100 GPUs, while pruning 13B and 27B-parameter models requires 4 H200 GPUs. Compared with Wanda and SparseGPT, this constitutes a much higher computational overhead—since pruning a 27B-parameter model using the above two methods only requires at most 1 A100 GPU.

**Questions:**

1. The authors used such a large number of GPUs to prune LLMs, so why not perform LoRA fine-tuning on the pruned model to obtain a better pruned model? It is suggested that the authors compare the accuracy, computational resources used, and time overhead for obtaining sparse LLMs between ELSA and the "Wanda + LoRA" approach.
2. Can the LLM sparsification technique used in this paper improve the model's inference speed? What advantages does it have compared with quantization techniques?

---

> ### Author Response · Authors · 2025-11-25
> **Response to reviewer KMnE (1/5)**
>
> We sincerely thank the reviewer for the encouraging remarks of our work as having *``**substantial theoretical contributions**’’*, and *``**exhibits significant performance advantages**’’*. We are also grateful for the constructive suggestions. While we have addressed specific comments below, please let us know if there is anything we need to address further. We would be keen to engage in any further discussion.

---

> > ### Author Response · Authors · 2025-11-25
> > **Response to reviewer KMnE (3/5)**
> >
> > **W2: Performance at moderate sparsity**
> > >  ``At low sparsity rates (50% and 60%), the accuracy of ELSA is significantly lower than that of the optimal L-ADMM baseline (Tables 7 and 8).’’
> >
> > Thank you for pointing this out. We had not been paying much attention to tune `ELSA` in the low sparsity regime. We found that our previous results were overtrained by mistake. We have addressed this and the updated results (average zero-shot accuracy) are as follows.
> >
> > | sparsity | dataset | L-ADMM | ALPS | SAFE | ELSA (before) | ELSA (after) |
> > |:--------:|:--------|-------:|-----:|-----:|--------------:|--------------:|
> > | 50%  | wiki    | 6.80   | 6.86 | 6.72 | 8.08          | 7.50 (-0.58) |
> > |          | c4      | 8.97   | 9.02 | 8.87 | 10.38         | 9.81 (-0.57) |
> > |          | avg_zeroshot  | 57.73  | 57.14| 55.30| 53.35         | 56.39 (+3.04) |
> > | 60%  | wiki    | 9.40   | 9.33 | 9.02 | 9.67          | 9.16 (-0.51) |
> > |          | c4      | 11.47  | 11.30| 11.40| 11.80         | 11.34 (-0.46) |
> > |          | avg_zeroshot  | 52.45  | 52.40| 50.57| 50.28         | 51.41 (+1.13) |
> >
> > As shown above, with the improved accuracy, `ELSA` becomes comparable to prior methods. We emphasize, however, that performance at high sparsity is much more critical than in the moderate sparsity, since the practical benefits of compression (i.e., inference acceleration, memory savings) only become significant at high sparsity levels [Gale+20, N+23, Zheng+22]. Thus, with respect, we believe that it is of more importance that `ELSA` has achieved a significant performance gap at high sparsity, rather than that of comparable performance in moderate sparsity.
> >
> > ---
> >
> > **W3: Performance gap between ELSA and dense LLMs**
> > > ``Although ELSA has advanced the performance of LLMs at high sparsity rates (70%-90%), there remains a substantial performance gap between ELSA and dense LLMs.’’
> >
> > While we agree that a gap to the dense model remains, we believe this work should still be regarded as a valid contribution: 1) ELSA achieves substantial improvements over existing pruning methods, and 2) prior work [Meng+24, Lu+25, Huang+25] consistently evaluates methods by their relative performance gains at the same sparsity level, especially at high sparsity rather than by perfectly matching the dense model.
> >
> > ---
> >
> > **W4: higher computational overhead**
> > > ``.. this constitutes a much higher computational overhead—since pruning a 27B-parameter model using the above two methods only requires at most 1 A100 GPU.’’
> >
> > To clarify, `ELSA` can also prune 7B models on a single A100 by using CPU offloading, just like `Wanda` and `SparseGPT`, so the memory setup in our experiments should not be seen as a strict requirement of `ELSA`. While this may increase the pruning time, pruning is a one-time procedure, whereas the dominant cost in practice is the repeated serving of the pruned model. Thus, what really matters is achieving high sparsity without sacrificing too much performance; since `ELSA` reaches much higher sparsity than competing methods, it can yield substantially larger cost savings at deployment.
> >
> > In other perspective, even if `SparseGPT` or `Wanda` were given substantially more compute resources (e.g., more data or additional training iterations), it would still be difficult for them to achieve the same level of sparsity as `ELSA` (refer to our global response). In other words, even if the pruning process is slightly faster, the 'effective efficiency' is compromised due to the significant disparity in the resulting sparsity.
> >
> > We believe our work makes a meaningful contribution by carefully re-examining the previously adopted problem setting and fundamentally addressing its limitations through a novel and effective approach. However, improving the cost efficiency of the pruning procedure itself is certainly a valuable future direction. We will make sure to incorporate this discussion appropriately in the final version of the paper.

---

> ### Author Response · Authors · 2025-11-25
> **Response to reviewer KMnE (2/5)**
>
> **W1: Difference between ELSA and other ADMM-based pruners**
> >  ``Both ALPS and L-ADMM have also proposed ADMM-based pruning algorithms for LLMs. The authors should clearly elaborate on the differences between ELSA and ALPS to highlight the unique contributions of ELSA’’
>
> The methodological differences between ELSA and existing ADMM-based approaches can be summarized as follows.
>
> | Method | O | M | Z | Q | G | F   |
> |-|-|-|-|-|-|-|
> | L-ADMM       | L-REM                   | Sequential layerwise | activation-aware      | x | gradual sparsity   | -   |
> | ALPS         | L-REM                   | Sequential layerwise | euclidean             | x | adaptive penalty   | PCG |
> | SAFE         | Sharpness-aware B-REM   | Sequential layerwise | activation-aware      | x | -                  | -   |
> | ELSA (Ours)  | NTP                     | holistic model-wide  | objective curvature-aware | o | -                  | -   |
>
> Here, each column corresponds to **Objective (O)**, **Model-wide procedure (M)**, **Z-minimization (Z)**, **Quantized state (Q)**, **Gradual constraint enforcement (G)**, **Final sparse fine-tuning (F)**.
>
> ***New objective and procedure with different purpose*** -- To be more precise, the purpose of each algorithm is different from its design.
>
> - **Difference-(O)**. Existing ADMM-based pruning methods adopt the same assumptions used in one-shot approaches (e.g., SparseGPT, Wanda), which were originally developed for memory-constrained settings. As a result, they adopt surrogate objectives such as L-REM / B-REM (Layerwise/Blockwise Reconstruction Error Minimization). Relying on the fact that such assumption may be too naive, `ELSA` was designed to solve for the original next token prediction (NTP) objective to overcome the limitations of LLM pruning and possibly find better solutions.
>
> - **Difference-(M)**. In particular, prior methods prune the entire model by simply applying locally defined methods in a disjoint, sequential manner, where we focus on the fact that this approach can have inherent limitations in finding a global solution, leading to suboptimal results.
>
> ***Advances in ADMM solver*** --  we further specialize and improve the ADMM solver to be effective for our objective.
>
> - **Difference-(Z)**. Unlike prior Euclidean or activation-aware projections used in the z-minimization step, which do not account for the surrogate-free objective, our method introduces a new projection mechanism that incorporates objective information (e.g., curvature) by directly using approximate derivatives.
> - **Difference-(Q)**. We also introduce a framework that quantizes the ADMM states to improve the scalability of `ELSA`. Particularly, quantizing the dual variable represents a nontrivial theoretical departure from prior methods.
> - For these reasons, we provide a new ADMM convergence analysis that explicitly accounts for the quantized dual updates.
>
> Thanks to this design, `ELSA` attains stable performance without relying on heuristic (G) or additional optimization procedures (F) . Of course, these orthogonal techniques could also be integrated into `ELSA`.
>
> ***Significantly improved results*** -- As a result, these differences collectively lead to substantial improvements in pruning performance and play a key role in surpassing the previously observed sparsity wall. Importantly, our ablation studies demonstrate that `ELSA`’s performance gains over prior methods do not stem from the common use of ADMM, but rather from the specific algorithmic differences we introduce.
>
> It is also worth noting that even seemingly small algorithmic differences have often led to substantial impact and are widely recognized as important contributions within the community. For example, ResNet [He+2016] differs from prior CNN architectures mainly by introducing skip connections, yet this change yielded dramatic performance improvements (and resulted in what is now the most cited paper of the 21st century to our best knowledge). Similarly, AdamW, the de facto optimizer for training Transformer-based architectures, may appear only marginally different from Adam in terms of simple algorithmic modifications, but its empirical advantages have been significant enough for it to replace Adam in practice. Along the same line, `ELSA` significantly differs from existing ADMM-based LLM pruning, achieving unprecedented performance gains. This establishes `ELSA` as a meaningful contribution, potentially driving a fundamental shift in approach for LLM pruning methods.
>
> We thank the reviewer for this valuable comment, and will add these points in the revised version of the paper.

---

> ### Author Response · Authors · 2025-11-25
> **Response to reviewer KMnE (4/5)**
>
> **Q1: Comparison with Wanda+LoRA**
> > ``.. why not perform LoRA fine-tuning on the pruned model to obtain a better pruned model? It is suggested that the authors compare the accuracy, computational resources used, and time overhead for obtaining sparse LLMs between ELSA and the "Wanda + LoRA" approach.’’
>
> We have conducted the requested experiment on LLaMA-2-7B, and the results are presented below. For fair comparison, both methods are trained with the same number of training tokens.
>
> | sparsity | method              | wiki$(\downarrow)$    | c4$(\downarrow)$     | avg_zero-shot $(\uparrow)$ |
> |:-------:|----------------------|-------:|------:|-------------:|
> |   0.90    | Wanda + lora         | 92.66   | 66.56 | 32.44 |
> |          | elsa             | **26.97** | **23.14** | **38.52** |
> |  0.95     | Wanda + lora         | 371.0   | 143.0 | 33.06 |
> |          | elsa             | **38.91** | **28.39** | **35.80** |
> |  0.99     | Wanda + lora         | 588.3   | 247.5 | 32.16 |
> |          | elsa             | **55.94** | **40.10** | **36.19** |
>
> As shown above, `ELSA` achieves substantially better performance than `Wanda + LoRA`, and this gap widens as the sparsity level increases (particularly in perplexity, the most direct metric for language modeling quality). This trend highlights the fundamental limitations of one-shot, local, and sequential pruning methods, which aligns exactly with prior findings that such local methods combined with fine-tuning tend to break down at high sparsity levels [Lee+20, Kurtic+23].
>
>
> In terms of computational resources, `Wanda+LoRA` and `ELSA` require 1 GPU and 4 GPUs, with wall-clock times of 3.99 hours and 2.14 hours respectively. If strict GPU constraints must be satisfied, one could apply CPU offloading (at the cost of runtime). However, regarding the fact that cost of the pruning procedure is less important than the inference cost (i.e., computational savings at deployment), `Wanda+LoRA` over `ELSA` diminishes the meaning of pruning since it suffers from inference overhead [Huang+25ICLR] and additional memory requirements for LoRA parameters.
>
> One point we would like to emphasize is that `ELSA` offers advantages not only in performance but also in terms of principled-ness. Heuristic, sequential approaches such as {one-shot pruning + fine-tuning} make it difficult to derive quantitative guarantees, such as bounds on the sub-optimality of the obtained sparse solution. In contrast, `ELSA` is designed directly from an optimization principle, and as demonstrated in Section 3.3, its convergence theory enables principled analyses of its effectiveness (e.g., deriving an upper bound on primal optimality through the weak duality gap).
>
> We thank the reviewer for this helpful suggestion; we will include this in the paper.
>
> ---
>
> **Q2: Inference speedup, quantization**
> > ``Can the LLM sparsification technique used in this paper improve the model's inference speed? What advantages does it have compared with quantization techniques?’’
>
> Thank you for the thoughtful question. `ELSA` can indeed improve inference speed. As higher sparsity generally leads to greater acceleration, `ELSA` can therefore deliver larger speedups than other pruning methods [Xia+23, Lu+25, Fan+25, Huang+25]. For instance, [Lu+25: Table 4] reports that end-to-end inference speedup increases as sparsity rises, reaching up to x1.71 (compared to dense) at 70% sparsity.
>
> `ELSA` is also compatible with semi-structured sparsity formats, which are known to be effective in inference speedup as reported in [Franstar&Alistarh+23, Sun+24,  Meng+24, Pool+21]. These formats were originally introduced by NVIDIA at the hardware level (e.g., Ampere architecture) to support acceleration of sparse models [Pool+21]. At that time, hardware support was limited to 2:4, 4:8 (50%) sparsity because models could not maintain accuracy beyond that level. By demonstrating the potential of high-sparsity LLMs, `ELSA` suggests that hardware optimized for more aggressive sparsity levels (e.g., around 75%) is both realistic and worth pursuing, and may help motivate such future designs..
>
> On the other hand, pruning and quantization should not be viewed as competing techniques that require a one-to-one comparison. Rather, they are complementary approaches. In fact, many prior studies have shown that combining pruning with quantization allows one to benefit from both techniques simultaneously [Han+16,Ye+19, Franstar&Alistarh+23,Yu+23, Park+22, Mozafarri+25].
> For this reason, `ELSA` is expected to be a promising method for achieving even faster inference than what quantization alone can offer, and exploring the synergy between the two approaches represents a natural and valuable avenue for future work.

---

> ### Author Response · Authors · 2025-11-25
> **Response to reviewer KMnE (5/5)**
>
> **Remark**
>
> We hope our response has addressed your comments reasonably well. If there is any remaining concern, please specify it for us so we can address them further during the discussion period. Otherwise, we find the overall rating of 4 to be quite inconsistent with the reviewer’s positive feedback (soundness=4/presentation=3/contribution=3 as well as the comments, e.g., “substantial theoretical contributions”, “ELSA exhibits significant performance advantages at high sparsity levels.”), and hence, we would greatly appreciate it if the reviewer could give it a reconsideration.
>
> ---
>
> **Reference**
>
> [Pool+21] Accelerating Inference with Sparsity Using the NVIDIA Ampere Architecture and NVIDIA TensorRT, Nvida blogpost \
> [Sutskever+13] Sutskever et al., On the importance of initialization and momentum in deep learning, ICML 2013 \
> [Frantar&Alistarh+23] Frantar et al., SparseGPT: Massive Language Models Can be Accurately Pruned in One-Shot, ICML 2023 \
> [Gale+20] Gale et al., Sparse gpu kernels for deep learning, SC20 2020 \
> [N+23] NVIDIA. cuSPARSE Library. https://docs.nvidia.com/cuda/cusparse/index. 2023 \
> [Zheng+22], Zheng et al., SparTA: Deep-Learning Model Sparsity via  Tensor-with-Sparsity-Attribute, OSDI 2022 \
> [Xia+23] Xia et al., Flash-LLM: Enabling Cost-Effective and Highly-Efficient Large
> Generative Model Inference with Unstructured Sparsity, VLDB 2023 \
> [Meng+24] Meng et la., ALPS: Improved Optimization for Highly Sparse One-Shot Pruning for Large Language Models, NeurIPS 2024 \
> [Huang+25] Huang et al., Determining Layer-wise Sparsity for Large Language Models Through a Theoretical Perspective, ICML 2025 \
> [Lu+25] Lu et al., Lua-LLM: Learning Unstructured-Sparsity Allocation for Large Language Models, NeurIPS 2025 \
> [Mozafarri+25] Mozafarri et al., When Quantization Isn’t Enough: Why 2:4 Sparsity Matters, Pytorch Blogpost https://pytorch.org/blog/when-quantization-isnt-enough-why-24-sparsity-matters/ \
> [Yu+23] Yu et al., Boost vision transformer with gpu-friendly sparsity and quantization, CVPR 2023 \
> [Park+22] Park et al., Quantized sparse training: a unified trainable framework for joint pruning and quantization in dnns, ACM Transactions on Embedded Computing System, 2022 \
> [Han+16] Han et al., DEEP COMPRESSION: COMPRESSING DEEP NEURAL
> NETWORKS WITH PRUNING, TRAINED QUANTIZATION AND HUFFMAN CODING, ICLR 2016 \
> [Ye+19] Ye et al., A Unified Framework of DNN Weight Pruning and Weight Clustering/Quantization Using ADMM, AAAI 19 \
> [Sun+24] A Simple and Effective Pruning Approach for Large Language Models, ICLR 2024

---

> ### Author Response · Authors · 2025-11-26
> **Updated results regarding Q2**
>
> We have updated results regarding inference speedup. `ELSA` with 90% sparsity achieves a **2.5x** inference speedup with **4.6x** memory savings compared to the dense model (LLaMA-2-7b). Please see the global response for full details.

---

### Official Review · Reviewer_v5xh · 2025-10-31

**Soundness:** 3
**Presentation:** 3
**Contribution:** 3
**Rating:** 6
**Confidence:** 4

**Summary:**

This paper introduces ELSA (Extreme LLM Sparsity via Surrogate-free ADMM), a method to prune large language models to extreme sparsity levels (up to 90\%) while preserving performance. It identifies limitations in existing pruning techniques, which rely on surrogate objectives like layer-wise reconstruction error minimization, leading to performance collapse beyond 50-60\% sparsity due to compounding errors and suboptimality. ELSA directly optimizes a sparsity-constrained problem using ADMM, incorporating objective-aware projections and avoiding surrogates. A quantized variant, ELSA-L, scales to 27B-parameter models with reduced memory. Experiments across models (OPT, Gemma, LLaMA) show ELSA achieves 5-30× lower perplexity and up to 6\% higher zero-shot accuracy at 90\% sparsity compared to baselines. Theoretical convergence is proven, highlighting potential for further LLM efficiency advancements.

**Strengths:**

1.	This paper is well-written and highly readable.

2.	At high sparsity rates, ELSA achieves significantly better accuracy than other baselines and successfully breaks through the performance limit of LLM sparsification.

3.	The experiments cover models of various parameter scales and different benchmarks, and conduct comprehensive evaluations of the LLMs' performance, including perplexity and zero-shot accuracy.

4.	The theoretical foundation is solid: convergence proofs for ELSA and ELSA-L are provided, based on standard assumptions (such as weak convexity and smoothness). This enhances the reliability of the method and aligns with the empirical results.

**Weaknesses:**

1.	ELSA's accuracy at low sparsity rates (50% and 60%) is lower than that of the baselines.

2.	It remains unclear how ELSA performs on larger models. Although the parameter scales of the models tested in the experiments range from 125 million to 27 billion, there is a lack of experimental results on even larger models, such as Llama-3-80B.

**Questions:**

1.	What is the computational efficiency of ELSA? How much time does it take to prune LLMs with different parameter scales?

2.	Can ELSA be used to prune Mixture-of-Experts (MoE) models?

---

> ### Author Response · Authors · 2025-11-25
> **Response to reviewer v5xh (1/2)**
>
> We sincerely appreciate the reviewer’s constructive feedback and recognition of our work’s ***clarity, solid theoretical foundations*** , and strong empirical results that  ***``breaks the performance limit of LLM sparsification’’***. We have addressed the reviewer’s specific comments below, and welcome any further suggestions that may improve the manuscript.
>
> ---
>
> **W1: Performance at moderate sparsity**
> >  `` `ELSA`'s accuracy at low sparsity rates (50% and 60%) is lower than that of the baselines.’’
>
>
> Thanks for pointing this out. We hadn’t been paying much attention to the low-sparsity regime, and it turned out we were actually overtraining by mistake. After correcting this, we reran the experiments, and the updated results are shown below.
>
> | sparsity | wanda | sparseGPT | L-ADMM | ALPS  | SAFE  | sparseLLM | ELSA (before) | ELSA (after) |
> |---------|-------|-----------|--------|-------|-------|-----------|----------------|---------------|
> | 50%     | 55.78 | 56.31     | 57.73  | 57.14 | 55.30 | 55.63     | 53.35         | 56.39 (+3.04) |
> | 60%     | 49.72 | 51.13     | 52.45  | 52.40 | 50.57 | 51.28     | 50.28         | 51.41 (+1.13) |
>
> As we can see above, `ELSA` achieves accuracy that is comparable to existing methods at moderate sparsity levels.
>
>
> However, we would like to emphasize that performance in the high-sparsity regime is far more crucial than that of moderate sparsity. This is obvious, since the practical benefits of sparsification such as inference acceleration and memory savings become significantly more pronounced as sparsity increases [Gale+20, N+23, Zheng+22]. Thus, we respectfully request the reviewer to recognize the significance of ELSA in its unique ability to scale effectively to high sparsity where its main strength lies rather than in the moderate-sparsity regime where ELSA maintains competitive performance.
>
>
>
> ---
>
> **W2: Experiments on larger models**
> > `` It remains unclear how ELSA performs on larger models…there is a lack of experimental results on even larger models, such as Llama-3-80B. ‘’
>
> While we understand the reviewer’s concern, running ELSA on such a scale (i.e., LLaMA-3-70B) would require at least 28x A100-80GB GPUs, which is unfortunately, not available in our research environment. We have also considered using commercial cloud platforms, but the cost is prohibitively high, making it realistically infeasible for us (e.g., over $160/hour for 4xEC2 p4 nodes in AWS). Therefore, we kindly request the reviewer’s generous consideration of these circumstances.
>
> Nevertheless, we are currently exploring whether it is possible to run experiments on LLaMA-3-70B using fully sharded data parallelism (FSDP) and CPU offloading [Rajbhandari+19]. Due to the nature of CPU offloading, such experiments will inevitably take substantial time, but if we manage to complete them, we will provide the results within the rebuttal period, or at the latest, include them in the camera-ready version.
>
> ---
>
> **Q1: Computational efficiency and wall-clock time**
> > `` What is the computational efficiency of ELSA? How much time does it take to prune LLMs with different parameter scales? ‘’
>
> Below are the measured wall-clock times for models of various sizes (x4 GPUs are used). Given that the evaluated models are large (up to 27B) and the pruning cost is incurred only once, we consider this to be reasonably affordable.
>
> | OPT-125m | OPT-1.3b | Gemma-2-2b | LLaMA-3.2-3b | LLaMA-2-7b | LLaMA-2-13b | Gemma-2-27b |
> |-----------|-----------|-----------|-----------|-----------|-|-|
> | 0.51H | 1.03H | 0.94H | 0.89H | 2.14H | 3.44H | 5.60H |
>
> To be more precise, the computational complexity of `ELSA` is $O(kLd^2)$, which can be contrasted with the layer-wise one-shot method `SparseGPT`, whose complexity is $O(L(Nd^2+d^3))$ (Here, k,L,d and N denote the number of forward/backward passes, the number of layers, the hidden dimension, and the number of calibration samples, respectively.
>
> However, comparing pruning complexity alone does not provide a meaningful assessment of ELSA’s overall efficiency. What ultimately matters is a method’s effective efficiency as well as the computational savings at deployment.
>
> In this work, we have demonstrated that `ELSA` achieves substantially higher performance than existing methods. We also observed that even if we allocate additional computational resources to local pruning methods, they still fail to match the performance level achieved by `ELSA` (see global response for details). In practice—especially for industrial applications and service providers where pruning is a one-time cost and the compressed model is used continuously—one must evaluate how effectively computational resources are converted into performance. From this perspective, `ELSA` delivering significantly improved performance, is far more advantageous in terms of effective efficiency.

---

> ### Author Response · Authors · 2025-11-25
> **Response to reviewer v5xh (2/2)**
>
> **Q2: Extension to MoE model**
> > ``Can ELSA be used to prune Mixture-of-Experts (MoE) models?’’
>
> Yes, `ELSA` can be applied to MoE pruning as it can be expressed in principle in the same constrained form handled by ELSA as follows:
> $\min_x f(x)\ \text{s.t. } \|x\|_0 \le d$.
>
> Although MoE includes non-continuous components such as routing and gating, prior work has established practical techniques that make MoE models trainable with standard GD-based optimizers [Jiang+24, Wei+24]. Thus, `ELSA`’s x-minimization step can be carried out in the same way. Moreover, expert-level pruning that is commonly considered in MoE can be incorporated simply by defining the sparsity constraint at the expert-selection level, allowing `ELSA` to be applied within the same constrained optimization framework without additional assumptions. Nevertheless, MoE has its own characteristics that affect its performance (e.g., expert capacity, gating load balancing) [Lopikhin+21, Fedus+22], and these should be considered carefully when extending `ELSA` to MoE to achieve its effectiveness. This is beyond the scope of our work, but we consider it a very promising direction for future research.
>
>
> ---
>
> **Remark**
>
> We would once again like to thank the reviewer for the positive evaluation of our work, including its clarity, theoretical grounding, and strong empirical performance (particularly at high sparsity), where `ELSA` provides clear advantages over prior methods. We hope that our responses have adequately addressed the reviewer’s concerns and that our clarifications help reinforce confidence in the contribution and significance of this work.
>
> ---
>
> **Reference**
>
> [Gale+20] Gale et al., Sparse gpu kernels for deep learning, SC20 2020 \
> [N+23] NVIDIA. cuSPARSE Library. https://docs.nvidia.com/cuda/cusparse/index. 2023 \
> [Zheng+22], Zheng et al., SparTA: Deep-Learning Model Sparsity via Tensor-with-Sparsity-Attribute, OSDI 2022 \
> [Rajbhandari+19] Rajbhandari et al., ZeRO: Memory Optimizations Toward Training Trillion Parameter Models, SC20 ‘2020 \
> [Lepikhin+21] Gshard: Scaling giant models with conditional computation and automatic sharding, ICLR ‘21 \
> [Fedus+22] Switch Transformers: Scaling to Trillion Parameter Models with Simple and Efficient Sparsity, JMLR ‘22 \
> [Jiang+24] Mixtral of Experts, arXiv ‘24 \
> [Wei+24] Skywork-MoE: A Deep Dive into Training Techniques for Mixture-of-Experts Language Models, arXiv ‘24

---

### Author Response · Authors · 2025-11-25
**Global response**

We thank the reviewers for their insightful feedback and positive assessment. We are encouraged by the unanimous recognition of `ELSA`’s *superior performance at high sparsity, solid theoretical guarantees, and the comprehensive evaluation across diverse models and benchmarks*. While we address specific comments in our individual responses, we use this global response to present additional improvements.

---

**Extremely high sparsity results**
In addition to our response to Reviewer `KMnE`, we further evaluated `ELSA` at extreme sparsity rates (95% and 99%) on LLaMA-2-7B. For fair comparison, all the methods are trained with the same number of training tokens.
| sparsity | method              | wiki    | c4     |
|:-:|-|-|-|
|   0.90       | Wanda + lora         | 92.66   | 66.56 |
|          | Wanda + full finetuning  | 42.40   | 34.87 |
|          | **ELSA**             | **26.97** | **23.14** |
|   0.95     | Wanda + lora         | 371.0   | 143.0 |
|          | Wanda + full finetuning  | 84.30   | 53.62 |
|          | **ELSA**             | **38.91** | **28.39** |
|    0.99      | Wanda + lora         | 588.3   | 247.5 |
|          | Wanda + full finetuning  | 146.37  | 71.64 |
|          | **ELSA**             | **55.94** | **40.10** |

As shown in the table, `ELSA` demonstrates robust performance even at 99% sparsity, outperforming Wanda+ full finetuning, which utilizes an equivalent computing budget with `ELSA`.

These results support the necessity of principled algorithms like `ELSA` for extreme sparsity, demonstrating clear superiority over simple heuristic methods such as `Wanda+LoRA/Full tuning`.

---

> ### Author Response · Authors · 2025-11-26
> **Global response (2)**
>
> **Inference acceleration / memory savings**
>
> In response to the questions raised by reviewer `KMnE`,`oqwk`, we performed an experiment to quantify the inference acceleration and memory savings achieved through ELSA's high sparsity.
>
> Our evaluation employed Macko [Macko+25], a very recent acceleration technique for sparse matrix-vector multiplication (SpMV) that also provides memory savings through its advanced sparse matrix format. Following the standard end-to-end experimental setup detailed in the Macko[Macko+25G] , we assessed LLaMA-2-7b at 50-90% sparsity, using a single consumer-grade RTX 3090 GPU.
>
> |              | dense | 50%           | 60%           | 70%           | 80%           | 90%            |
> |--------------|--------|---------------|---------------|---------------|---------------|----------------|
> | *latency (s) | 1.84   | 1.37 (×1.34)  | 1.14 (×1.61)  | 0.95 (×1.94)  | 0.75 (×2.45)  | **0.72 (×2.50)**   |
> | tokens/s     | 54.47  | 72.81 (×1.33) | 87.50 (×1.61) | 104.89 (×1.93)| 133.41 (×2.45)| **139.47 (×2.56)** |
> | memory (MB)  | 13596  | 8870 (×1.53)  | 7148 (×1.90)  | 5603 (×2.42)  | 4064 (×3.35)  | **2918 (×4.60)**   |
>
> \* end-to-end latency for text generation.
>
> The clear benefit of increased sparsity is evident in the enhanced inference acceleration and memory savings observed. Notably, from a sparsity level of 70% onwards, these gains become more significant, achieving up to a **2.5x** increase in inference speed and a **4.6x** compression rate when compared to their dense counterparts.
>
> These observations highlight an important point: high-sparsity regimes are precisely where meaningful acceleration and memory savings emerge, and `ELSA` demonstrates that LLMs can be pushed into this regime while still maintaining strong performance. We believe this not only addresses the reviewers’ concerns but also underscores `ELSA`’s contribution — it helps make high-sparsity, deployment-oriented configurations practically attainable, enabling sparsity-driven optimizations that were difficult to realize with existing pruning methods. We thank reviewers `KMnE` and `oqwk` again for this valuable comment and will include these results in the updated version of the paper.
>
> ---
>
> **Reference**
>
> [Macko+25] Macko et al., MACKO: Sparse Matrix-Vector Multiplication for Low Sparsity, arXiv 2025 \
> [Macko+25G] Macko github, https://github.com/vlejd/macko_spmv/blob/master/TECHNICAL_README.md

---

> > ### Author Response · Authors · 2025-11-29
> > **Global response (3)**
> >
> > **Inference acceleration/memory savings for extreme sparsity**
> >
> > Following our prior global response, we performed additional experiments at the same setting to further evaluate the inference accelerations and memory savings achieved through extreme sparsity. The results are presented below (LLaMA-2-7b).
> > |               | dense | 90%             | 95%              | 99%              |
> > |---------------|-------|-----------------|------------------|------------------|
> > | *latency (s)  | 1.84  | 0.72 (x2.5)     | 0.46 (x4)        | 0.63 (x2.92)     |
> > | tokens/s      | 54.47 | 139.47 (x2.56)  | 216.98 (x3.98)   | 159.56 (x2.92)   |
> > | memory (MB)   | 13596 | 2918 (x4.6)     | 1743 (x7.8)      | 1787 (x7.6)      |
> >
> > *end-to-end latency for text generation.
> >
> > Compared to the dense model, a 95% sparsity level yields up to **4×** faster inference and **7.8×** lower memory usage, underscoring the practical potential of ELSA at extreme sparsity.

---

### Meta-Review · Area_Chair_DyG2 · 2025-12-14

**Summary:**

The reviewers’ discussion focused on whether ELSA meaningfully advances large-language-model pruning beyond existing methods, particularly in the high-sparsity regime, and on the practical implications of extreme sparsity in terms of performance, efficiency, and deployment. Reviewers broadly agreed that the paper is technically sound, theoretically grounded, and empirically strong at high sparsity levels (70–99%), where existing pruning approaches typically fail. Divergence arose around performance at moderate sparsity, computational overhead during pruning, and whether sparsity gains translate into real inference acceleration and memory savings.

In response, the authors provided extensive additional experiments and clarifications. These include corrected results at moderate sparsity, new comparisons against Wanda+LoRA, additional analyses of inference speedup and memory savings using recent sparse inference engines, and detailed discussion of computational cost, scalability, and applicability to settings such as MoE models. These additions directly address the main points raised in the reviews.

**Reviewer Concerns:**

Addressed

* Concerns about reduced accuracy at moderate sparsity were addressed by identifying and correcting an overtraining issue
* Questions regarding the distinction between ELSA and prior methods were addressed through a detailed methodological comparison
* Requests to compare ELSA against Wanda+LoRA were addressed with new experiments demonstrating substantially better performance for ELSA
* Concerns about the lack of practical metrics were addressed by adding explicit measurements of inference latency, throughput, and memory usage
* Questions regarding applicability to MoE models were addressed conceptually

Unresolved

* A performance gap between pruned models and dense models remains at very high sparsity, although this was acknowledged and framed as consistent with prior literature.
* ELSA’s pruning procedure incurs higher computational overhead than simpler one-shot methods, which remains a tradeoff despite arguments that pruning is a one-time cost.
* Some aspects of the method remain inherent limitations, though partially mitigated by empirical robustness and discussion.

**Reviewer Scores:**

* Reviewer v5xh: Likely unchanged or slightly higher, given that concerns about moderate sparsity, computational efficiency, and scalability were directly addressed with corrected results and additional measurements.
* Reviewer KMnE: Likely modestly higher, as detailed distinctions from prior ADMM methods, new Wanda+LoRA comparisons, and added inference-speed analyses directly respond to the reviewer’s main weaknesses and questions.
* Reviewer oqwk: Likely unchanged or slightly higher, since practical deployment considerations, inference acceleration, memory savings, and compute---performance tradeoffs were explicitly quantified in follow-up experiments.

---

### Decision · Program_Chairs · 2026-01-26

Accept (Poster)